# Divisive Normalization Shapes Low-Rank Slow Manifolds for Continuous Working Memory

## Abstract

The ability to robustly maintain and update continuous variables is a hallmark of working memory. While classical continuous attractor networks suffer from severe fine-tuning fragility, standard artificial recurrent neural networks (RNNs) like GRUs and LSTMs typically fail to stably learn continuous manifolds, instead shattering the state space into discretized point attractors. To bridge this gap, we draw inspiration from divisive normalization, a canonical neural computation widely observed across cortical circuits, and propose the Recurrent Divisive Normalization Network (RDNN), a biologically motivated architecture that replaces standard additive gating with a dynamic, multiplicative divisive bottleneck. Through dynamical systems analysis on canonical working memory tasks, we demonstrate that the RDNN naturally converges to robust, high-fidelity slow manifolds. Furthermore, we mathematically and empirically reveal that divisive normalization acts as a powerful activity-dependent implicit regularizer during Backpropagation Through Time (BPTT). This mechanism induces an efficient self-compression of the network's effective rank, confining the recurrent dynamics to a tight, low-dimensional subspace while avoiding the optimization pathologies associated with explicit low-rank factorization. Finally, ablations demonstrate that while subtractive inhibition can maintain static memories, divisive normalization is mathematically essential to prevent manifold shattering under time-varying inputs. Our findings identify divisive normalization not merely as a biological artifact, but as a critical computational mechanism for learning high-fidelity continuous representations.

## 1 Introduction

The ability to maintain and manipulate continuous variables over time, such as spatial location, head direction, or accumulated evidence, is a hallmark of biological working memory. In computational neuroscience, this cognitive function is classically modeled using Continuous Attractor Networks (CANs) (Wimmer et al., 2014; Khona & Fiete, 2022). However, CANs are notoriously brittle: they suffer from the *fine-tuning problem*, where infinitesimal perturbations in recurrent dynamics can destroy the continuous manifold of equilibria, causing the stored memory to rapidly drift or collapse (Seung, 1996; Koulakov et al., 2002). In parallel, while standard artificial Recurrent Neural Networks (RNNs) like LSTMs and GRUs excel at sequential tasks, recent dynamical systems analyses reveal that they struggle to stably learn true continuous attractors. Instead, they often discretize the continuous state space, shattering the manifold into localized point attractors separated by saddles (Jordan et al., 2021). This raises a fundamental question: What computational mechanisms allow biological neural networks to robustly represent continuous working memory, and how can we integrate them into learnable computational models?

To bridge this gap, we draw inspiration from divisive normalization (DN), a canonical neural computation widely observed across cortical circuits for dynamic gain control and contrast invariance (Carandini & Heeger, 2012). While previous works have explored DN for sensory processing or unconditional stability in hand-crafted circuits (Heeger & Mackey, 2019; Rawat et al., 2024), its role in shaping the optimization and topological landscape of learned working memory remains largely unexplored. In this work, we propose the Recurrent Divisive Normalization Network (RDNN), a biologically motivated RNN architecture that replaces standard additive gating with a dynamic, multiplicative divisive bottleneck. By training the RDNN on

canonical continuous working memory tasks, including autonomous maintenance (memory-guided saccade) and input-driven updating (angular velocity integration), we investigate how this structural constraint shapes the learned neural representations.

Through dynamical systems analysis, we uncover that divisive normalization is not merely a biological artifact, but a critical computational mechanism for high-fidelity continuous memory. Furthermore, this mechanism induces a significant *self-compression* of the network's effective rank, forcing the recurrent dynamics into a tight, low-dimensional subspace without requiring explicit low-rank parameterization.

Our main contributions are summarized as follows:

- **Biologically-inspired Architecture for Continuous Memory:** We introduce the RDNN and demonstrate that it learns robust slow manifolds for continuous working memory, overcoming the topological discretization inherent in standard GRU and LSTM architectures.

- **Emergent Low-Rank Dynamics:** We mathematically and empirically show that divisive normalization acts as a strong implicit low-rank regularizer. The RDNN yields significantly lower effective ranks than baselines, and avoids the non-convex optimization anomalies associated with explicit low-rank factorization.

- **Mechanistic Ablation of Inhibition:** Through systematic ablation, we delineate the functional niches of divisive versus subtractive inhibition. We demonstrate that while subtractive inhibition can maintain static memories, multiplicative gain control is essential to prevent manifold shattering under time-varying external inputs.

## 2 Related Works

**Divisive Normalization**  In neuroscience, divisive normalization is a canonical cortical operation that implements gain control and robust encoding. Classical studies (Heeger, 1992; Carandini & Heeger, 2012) demonstrated that V1 neurons divide their inputs by a pooled activity, explaining nonlinear contrast effects. Recent work has quantitatively fit DN models to neural data (Sawada & Petrov, 2017) and even learned orientation-specific normalization kernels from visual cortex (Burg et al., 2021). In parallel, machine learning uses related normalization layers (Ioffe & Szegedy, 2015; Xu et al., 2019), but the neuroscience view emphasizes explicit dynamic divisive feedback. These findings underscore that normalization circuits filter out noise and shape responses before further processing.

Several recent models incorporate DN into recurrent memory circuits. Heeger & Mackey (2019) introduced ORGaNICs (Oscillatory Recurrent Gated Neural Integrator Circuits), a biologically plausible RNN framework with dynamic DN that unifies sensory gain control, gated integration, and working-memory maintenance. Subsequently, Rawat et al. (2024) analyzed these circuits to prove their unconditional Lyapunov stability under broad conditions, showing they can be trained with backpropagation without special tricks and perform comparably to LSTMs on sequential tasks. Together, these works suggest that adding divisive feedback enables noise suppression and stable delay-period activity, unifying sensory gain control with working-memory maintenance.

**Continuous Attractor Networks and Slow Manifolds**  A classic model for analog memory is the continuous attractor network (Khona & Fiete, 2022; Wimmer et al., 2014), which can store graded values indefinitely. Such models have been used to explain spatial working memory in prefrontal cortex, for example by linking the precision of recall to the width of a bump attractor (Wimmer et al., 2014). However, continuous attractors suffer from a severe structural instability known as the *fine-tuning problem* (Seung, 1996; Koulakov et al., 2002): even infinitesimal perturbations in recurrent dynamics can destroy the continuous manifold of equilibria, causing the stored representation to rapidly drift or collapse.

To address this fragility, recent recurrent network analyses have extended these ideas. Ghazizadeh & Ching (2021) optimized many RNNs on a memory task and found that a prevalent solution is to encode memories along slow stable manifolds rather than fixed-point attractors; this leads to phasic activity that gradually

drifts but is more robust to noise. Likewise, Ságodi et al. (2024) show that networks trained on analog tasks exhibit approximate continuous attractors: even though exact continuous attractors are fragile to perturbations, a persistent low-dimensional slow manifold survives most bifurcations. These studies indicate that flexible working-memory codes can emerge without fine-tuned attractors. Meanwhile, dynamical analyses of standard gated architectures, such as Gated Recurrent Units (GRUs) viewed through continuous-time systems, have highlighted their structural inability to stably learn or represent continuous attractors (Jordan et al., 2021), further underscoring the need for alternative biologically inspired architectures with multiplicative feedback.

**Low-rank and low-dimensional RNNs**  A related line of work studies how structured connectivity produces low-dimensional dynamics. Mastrogiuseppe & Ostojic (2018) introduced the low-rank RNN framework: if recurrent weights have a low-dimensional component plus noise, the network's dynamics are confined to a low-dimensional subspace determined by that structure. They show one can design minimal connectivity to implement particular computations (e.g. integration or oscillation). More recently, Mastrogiuseppe et al. (2025) extended this to stochastic inputs: they prove that low-dimensional inputs keep the activity low-rank, but high-dimensional noise can inflate the network's activity dimension beyond the nominal rank. Bridging local constraints with global behavior, Shao et al. (2025) demonstrated that specific local synaptic patterns in excitatory-inhibitory networks mathematically induce low-rank structures that dominate the overall recurrent dynamics. In general, tools like fixed-point analysis (Sussillo & Barak, 2013) reveal that trained RNNs operate via a few fixed points and slow trajectories. These insights explain how E-I networks with normalization or other constraints could generate the simple low-dimensional manifolds observed in memory tasks.

## 3  Proposed Model

To investigate how biological constraints shape the neural representations of continuous working memory, we draw inspiration from divisive normalization, a common mechanism for dynamic gain control in the brain. As illustrated in Figure 1**A**, the proposed Recurrent Divisive Normalization Network (RDNN) consists of two interacting populations: a principal excitatory population $\mathbf{R} \in \mathbb{R}_+^H$ (representing the working memory state) and an auxiliary inhibitory pool $\mathbf{G} \in \mathbb{R}_+^H$ (providing dynamic gain control), where $H$ is the hidden dimension.

The continuous-time dynamics of the RDNN are governed by a system of coupled stochastic differential equations (SDEs). To emulate the intrinsic stochasticity and synaptic noise characteristic of biological neural systems (Faisal et al., 2008), we inject state noise into the dynamics. The dynamics of the network is defined as:

$$
\begin{aligned}
\tau_{\mathbf{R}} d\mathbf{R} &= \left( -\mathbf{R} + \frac{\mathbf{J}f(\mathbf{R}) + \mathbf{I}(t)}{\boldsymbol{\eta} + \mathbf{G}} \right) dt + \sigma_{\mathbf{R}} d\mathbf{W_R}, \\
\tau_{\mathbf{G}} d\mathbf{G} &= \left( -\mathbf{G} + \mathbf{W}f(\mathbf{R}) \right) dt + \sigma_{\mathbf{G}} d\mathbf{W_G}.
\end{aligned}
\tag{1}
$$

Here, $\mathbf{I}(t) \in \mathbb{R}^H$ denotes the external input stimulus at time $t$; $f(\cdot)$ is an elementwise nonlinear activation function (e.g., ReLU or Tanh) that maps the state variables to firing rates; $\mathbf{J} \in \mathbb{R}_+^{H \times H}$ is the recurrent excitatory weight matrix (see the arrows between $R_i$ in Figure 1**A**), and $\mathbf{W} \in \mathbb{R}_+^{H \times H}$ is the projection weight from the excitatory to the inhibitory population (the arrows from $R_i$ to $G_i$ in Figure 1**A**). The division operation $\frac{(\cdot)}{(\cdot)}$ is performed elementwise and represents the shunting inhibition exerted by $\mathbf{G}$ on $\mathbf{R}$ (square-headed connections in Figure 1**A**); $\boldsymbol{\eta} \in \mathbb{R}_+^H$ is a semi-saturation constant vector that prevents division by zero and sets the neurons' baseline responsiveness; $\tau_{\mathbf{R}}, \tau_{\mathbf{G}} \in \mathbb{R}_+^H$ are the membrane time constants of the two populations; and $d\mathbf{W_R}$ and $d\mathbf{W_G}$ are independent standard Wiener processes scaled by the noise intensities $\sigma_{\mathbf{R}}$ and $\sigma_{\mathbf{G}}$, respectively, which model the state noise. In our model, parameters $\mathbf{J}, \mathbf{W}, \boldsymbol{\eta}, \tau_{\mathbf{R}}, \tau_{\mathbf{G}}$ are all learnable and constrained to be positive, ensuring that the network operates as a positive dynamical system consistent with biological principles such as Dale's law (Fitzpatrick et al., 1987) (see Appendix B).

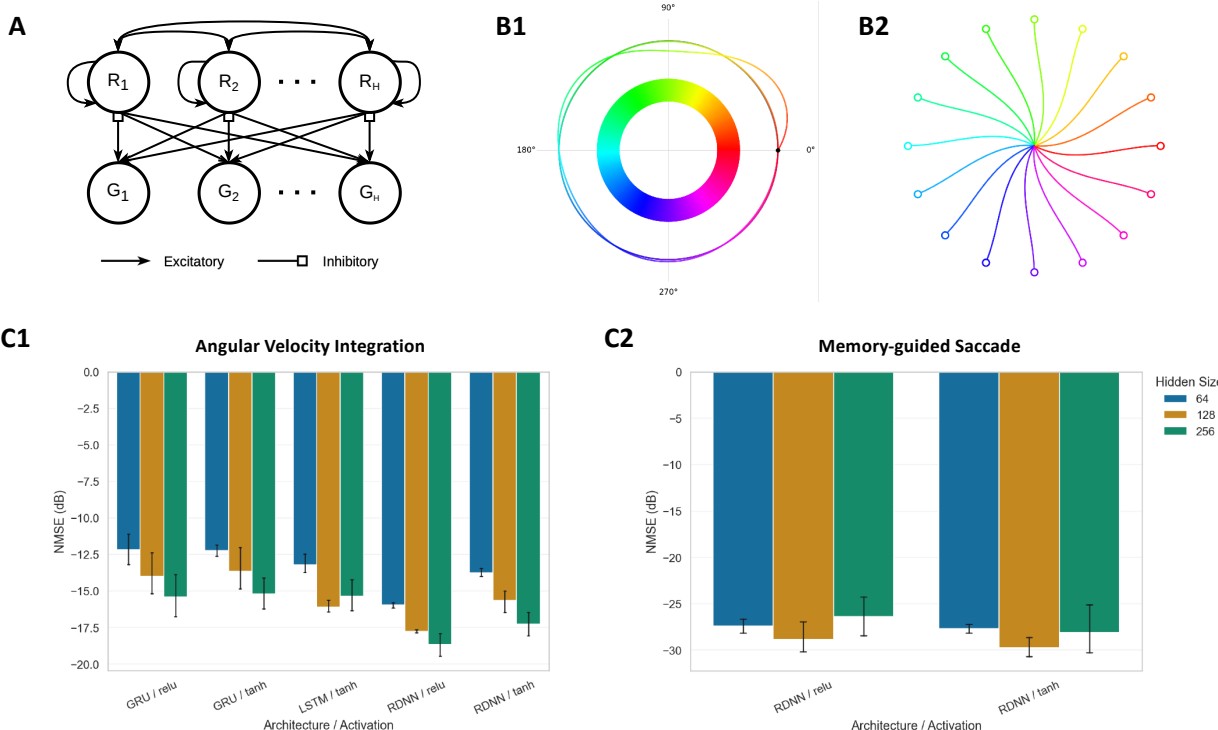

Figure 1: Overview of the proposed RDNN and its performance on continuous working memory tasks. **A** The RDNN architecture, featuring an excitatory population **R** and an inhibitory pool **G** for divisive gain control (square-headed lines). **B** Schematics of the continuous working memory tasks: angular velocity integration (B1) and memory-guided saccade (B2), with colors indicating the encoded angular variable. **C** Test NMSE (dB) across different architectures, activations, and hidden sizes. Error bars denote the standard deviation across different random seeds.

Unlike standard RNNs that rely on gating mechanisms, the RDNN employs a divisive bottleneck. The inhibitory state **G** dynamically tracks the overall network activity via $\mathbf{W}f(\mathbf{R})$. When the recurrent drive $\mathbf{J}f(\mathbf{R})$ or external input $\mathbf{I}(t)$ fluctuates, **G** proportionally scales the denominator, thereby stabilizing the network against explosive positive feedback while preserving the relative activation patterns. As we will demonstrate, this structural inductive bias naturally induces low-rank, high-fidelity slow manifolds.

## 4 Slow Manifold Dynamics in Continuous Working Memory Tasks

To understand the computational mechanisms underlying continuous working memory, we trained the proposed RDNN and standard gated models (GRU, LSTM) across various hidden sizes (64, 128, 256) and activation functions (ReLU, Tanh) on two canonical tasks (Figure 1**B**): Angular Velocity Integration (input-driven updating) and Memory-guided Saccade (autonomous maintenance). While the RDNN achieves comparable or superior task performance to the baselines (Figure 1**C**), we seek to uncover the fundamental differences in their learned representations. To this end, we systematically analyze their slow manifold dynamics, focusing on the topological distribution of fixed points, eigenvalue spectra, uniform flow norms, and manifold reliability (see Appendix D for detailed analytical methods). Notably, as detailed in Appendix C, both GRU and LSTM failed to converge on the memory-guided saccade task, and the ReLU-LSTM variant was untrainable due to gradient instability. This failure reflects their structural bias toward discrete attractors, which may cause catastrophic state drift during prolonged zero-input delays (Jordan et al., 2021). Consequently, our analysis for the memory-guided saccade task focuses exclusively on the RDNN.

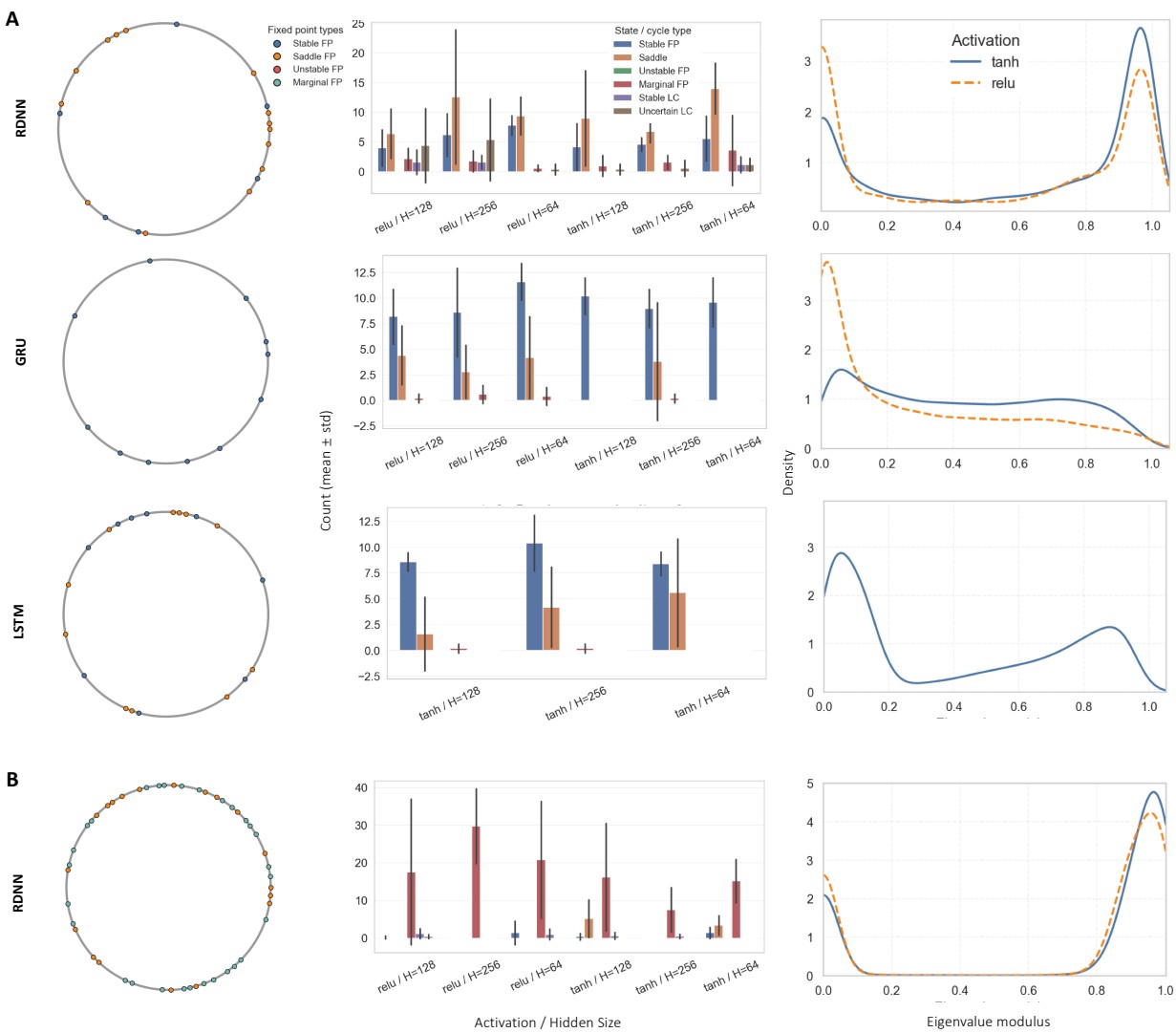

Figure 2: Dynamical topology and spectral properties of the learned slow manifolds. **A** Angular Velocity Integration. Left: Representative fixed-point distributions on the output ring. Middle: Statistical counts of topological states across random seeds. Right: Eigenvalue modulus density. RDNN shows a structured bimodal distribution indicating normal hyperbolicity, whereas baselines exhibit diffuse spectra without a clear gap. **B** Memory-guided Saccade (RDNN only). In this autonomous maintenance task, the RDNN manifold is overwhelmingly dominated by marginal fixed points (left, middle), approximating a theoretical continuous attractor. The corresponding eigenvalue spectrum (right) displays a strict spectral gap, indicating near-perfect time-scale separation.

## 4.1 Input-Driven Integration: Rotational Dynamics vs. Discretized States

The angular velocity integration task requires the network to continuously rotate its internal state in response to external inputs. Here, the topological strategies diverge significantly between architectures (Figure 2**A**, middle). GRU and LSTM networks predominantly form alternating discrete stable fixed points and saddle points. This suggests that gated RNNs effectively discretize the continuous variable, forcing the state to jump between localized basins of attraction (Ghazizadeh & Ching, 2021).

In contrast, the RDNN exhibits a more fluid topological structure. Its state space is frequently dominated by saddle points outnumbering stable fixed points, supplemented by marginal fixed points and limit cycles, indicating a flatter energy landscape operating near a bifurcation, e.g., saddle-node on an invariant circle (Strogatz, 2018; Seung, 1996; Maheswaranathan et al., 2019). Rather than acting as strict barriers, these saddles and marginal points function as *slow channels* (Strogatz, 2018; Ermentrout & Kopell, 1986), or *stable heteroclinic channels* (Rabinovich et al., 2008). This fluid topology allows external inputs to smoothly drive the state along the manifold without being trapped in deep localized attractors (Khona & Fiete, 2022).

This distinction is further illuminated by the complex eigenvalue spectrum of the Jacobians (Figure A5). The RDNN displays a highly structured, tripartite spectrum: the majority of eigenvalues cluster strictly along the real axis, while the remaining modes form off-axis conjugate pairs spanning the interior, alongside a distinct set of boundary conjugate pairs near the unit circle ($|z| \approx 1, \text{Im}(z) \neq 0$). These boundary conjugate pairs provide the marginally stable rotational dynamics required for continuous integration (Mastrogiuseppe & Ostojic, 2018; Sussillo & Barak, 2013), while the interior conjugate pairs govern damped transient oscillations that stabilize state transitions along the manifold (Hennequin et al., 2014). Conversely, the spectra of GRU and LSTM resemble a diffuse cloud, lacking both a clear spectral gap and structured rotational modes (Rajan & Abbott, 2006; Farrell et al., 2022). Consequently, the RDNN maintains a significantly lower uniform norm than the baselines (Figure A6, left), reflecting a smoother manifold that is less susceptible to input-induced drift.

## 4.2 Autonomous Maintenance: Emergence of Approximate Continuous Attractors

In the autonomous memory-guided saccade task, the RDNN exhibits dynamics that closely approximate a theoretical continuous attractor. Topological classification of the identified slow manifolds (Figure 2**B**, middle) reveals that the RDNN's state space is overwhelmingly dominated by marginal fixed points. This neutral stability along the manifold is a defining characteristic of continuous attractors (Seung, 1996; Khona & Fiete, 2022), allowing the network to maintain arbitrary continuous values without drifting toward discrete attractors (Compte, 2000; Koulakov et al., 2002).

This near-critical dynamic regime is quantitatively supported by the uniform norm of the vector field restricted to the manifold (Figure A6, right). The RDNN achieves an exceptionally low uniform norm (on the order of $10^{-3}$ to $10^{-4}$), indicating an extremely flat energy landscape. Instance-level analysis (Figure A8) further confirms that this continuous attractor-like topology is consistently learned across different random seeds and hidden sizes, demonstrating that divisive normalization provides a strong inductive bias for high-fidelity autonomous memory.

## 4.3 Spectral Properties, Activations, and Manifold Reliability

The eigenvalue modulus density (Figure 2**A** and 2**B**, right) highlights the normal hyperbolicity of the RDNN (Fenichel, 1979; Chaudhuri et al., 2019), though its strictness depends inherently on the task regime. In the autonomous memory-guided saccade task, the RDNN exhibits a sharp bimodal distribution with a strict spectral gap between peaks near 0 (fast dissipation) and 1 (slow maintenance). As derived in Appendix E.3, this separation is actively sculpted by BPTT and noise-induced regularization, catalyzed by the dynamic gain control of divisive scaling. However, in the input-driven angular velocity integration task, this gap is partially filled with intermediate eigenvalues. This non-zero density is a functional necessity: it reflects the intermediate time-scale modes required to continuously process external velocity inputs and couple them to the slow manifold (Bondanelli & Ostojic, 2020; Hennequin et al., 2014).

Interestingly, the choice of activation function subtly modulates this spectrum during active integration. Specifically in the angular velocity integration task, the ReLU activation in the RDNN produces a significantly higher peak at 0 compared to Tanh. This suggests that ReLU's hard-thresholding, when stabilized by divisive normalization, induces a sparser representation during dynamic updating (Glorot et al., 2011). Strictly silent neurons provide zero gradients, thereby instantly annihilating orthogonal noise and creating a *harder* manifold during continuous integration (Pennington et al., 2017; Engelken et al., 2023; Laurent & von Brecht, 2017; Ahmad & Hawkins, 2016).

Finally, this structured dynamical bias translates directly to training reliability. As shown in Figure A4, the RDNN successfully forms reliable, topologically correct ring manifolds in approximately 90% of the trained instances. In contrast, GRU and LSTM models yield reliable manifolds in less than 50% of cases, further underscoring the necessity of divisive normalization for robust continuous working memory.

## 5 Low-Rank Analysis

The algebraic rank of the recurrent connectivity matrix fundamentally bounds and shapes the dimensionality of the learned state-space trajectories in dynamical systems theory (Mastrogiuseppe & Ostojic, 2018). To understand whether the low-dimensional representations discovered before are a fundamental property of divisive normalization or merely a coincidental feature of training, we conduct a comprehensive low-rank analysis. We first examine the unconstrained effective rank of different architectures. Motivated by the naturally emergent low-rank structure of the RDNN, we then investigate the effects of forcing an explicit low-rank bottleneck by training an $H = 128$ RDNN with explicitly factorized recurrent weights, $\mathbf{J} = \mathbf{J}_1\mathbf{J}_2$ and $\mathbf{W} = \mathbf{W}_1\mathbf{W}_2$, across varying bottleneck ranks $r \in \{8, 16, 24, 32\}$.

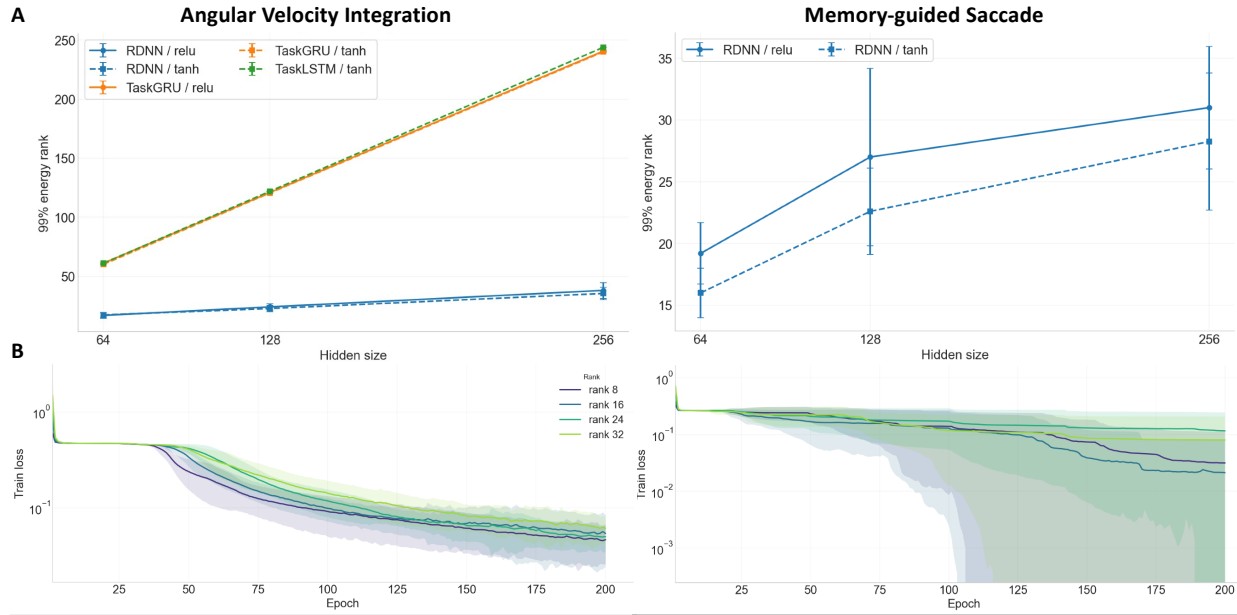

Figure 3: Effective rank scaling and explicit low-rank training dynamics. **A** The 99% energy effective rank as a function of physical hidden size ($H$). Across both tasks, the RDNN maintains an exceptionally compact, sub-linear scaling, whereas standard gated baselines (GRU, LSTM) scale linearly and exploit nearly the entire physical state space (GRU and LSTM are absent in the memory-guided saccade task due to non-convergence). **B** Training convergence curves (Loss vs. Epoch) of the explicitly factorized low-rank RDNN ($H = 128$) across bottleneck ranks $r \in \{8, 16, 24, 32\}$. The memory-guided saccade task (right) illustrates the higher-rank convergence anomaly, where wider explicit bottlenecks (e.g., $r = 32$) suffer from wider optimization variance and slower descent compared to tighter bottlenecks (e.g., $r = 8$). Error bands and error bars represent the standard deviation across random seeds.

### 5.1 Effective Rank Scaling and the Self-Compression Phenomenon

We first compare the 99% energy effective rank of the unconstrained models across different hidden sizes (Figure 3**A** and Table 1). While the effective ranks of the GRU and LSTM baselines scale linearly with the physical hidden size $H$ (essentially exploiting all available degrees of freedom), the RDNN exhibits a highly sub-linear, flat scaling. For instance, at $H = 256$ in the integration task, the RDNN maintains an effective

Table 1: The 99% energy effective rank (mean $\pm$ std over different random seeds) across different network architectures, activation functions, and hidden sizes ($H$).

| Model | Activation | Angular Velocity Integration | | | Memory-guided Saccade | | |
|---|---|---|---|---|---|---|---|
| | | $H = 64$ | $H = 128$ | $H = 256$ | $H = 64$ | $H = 128$ | $H = 256$ |
| **RDNN (Ours)** | **relu** | **$16.8 \pm 2.4$** | **$24.0 \pm 2.5$** | **$38.0 \pm 6.7$** | **$19.2 \pm 2.5$** | **$27.0 \pm 7.2$** | **$31.0 \pm 5.0$** |
| | **tanh** | **$17.2 \pm 1.9$** | **$22.6 \pm 2.6$** | **$35.4 \pm 5.3$** | **$16.0 \pm 2.0$** | **$22.6 \pm 3.5$** | **$28.3 \pm 5.6$** |
| GRU | relu | $61.0 \pm 0.0$ | $121.0 \pm 0.0$ | $240.8 \pm 0.4$ | — | — | — |
| | tanh | $60.0 \pm 0.0$ | $120.6 \pm 0.5$ | $240.2 \pm 0.4$ | — | — | — |
| LSTM | tanh | $61.0 \pm 0.0$ | $122.0 \pm 0.0$ | $244.0 \pm 0.0$ | — | — | — |

rank of approximately $35.4 \sim 38.0$, whereas the baselines require a rank of over 240. Mathematically, this emergent low-rank phenomenon is driven by the inverse gradient scaling of the divisive term, which acts as an activity-dependent implicit regularizer during backpropagation (see Appendix E.2 and E.4).

When we enforce an explicit bottleneck of rank $r$ on the $H = 128$ RDNN, a self-compression phenomenon emerges during training. As illustrated in Figure A10, regardless of how wide we set the explicit bottleneck $r \in \{8, 16, 24, 32\}$, the trained networks consistently compress their effective ranks back to a much lower, intrinsic dimensionality: approximately $3.5 \sim 4.0$ for the integration task, and $\sim 2.0$ for the saccade task. This extreme self-compression indicates that the intrinsic dimensionality of the slow manifold for these continuous tasks is fundamentally low. Intriguingly, the effective rank of these explicit low-rank networks is significantly lower than that of the unconstrained RDNN, and this hyper-compression is mathematically driven by the coupled BPTT gradient updates in the factorized parameter space, which accelerate singular value decay and act as a powerful multiplier to the implicit divisive regularizer (see Appendix E.5.2).

## 5.2 The Higher-Rank Convergence Anomaly

A key feature of low-rank parameterization is its impact on the optimization landscape. Counter-intuitively, our training curves (Figure 3**B**) reveal a higher-rank convergence anomaly: in many instances, networks with higher explicit ranks (e.g., $r = 32$) are significantly more difficult to optimize and converge more slowly than those with lower explicit ranks (e.g., $r = 8$). This is particularly evident in the memory-guided saccade task (Figure 3**B**, right), where the $r = 32$ configurations exhibit a massive variance in training loss across seeds and suffer from prolonged convergence plateaus, whereas the $r = 8$ networks converge more rapidly and stably.

As derived in Appendix E.5.3, this anomaly can be understood through the lens of non-convex matrix factorization. A wider bottleneck ($r = 32$) increases the dimensionality of the search space, introducing a vast number of redundant scaling symmetries, flat saddle points, and degenerate directions in the product space $\mathbf{J}_1 \mathbf{J}_2$ (Li et al., 2019; Valavi et al., 2020). Conversely, a tight bottleneck ($r = 8$) acts as a strong implicit regularizer, smoothing the optimization landscape by restricting the gradient search strictly within the low-dimensional subspace more relevant to the continuous attractor (Arora et al., 2019). In contrast, the unconstrained RDNN is parameterized directly without product-form weights. This flat parameterization avoids these non-convex scaling symmetries; as a result, increasing the hidden size $H$ of the unconstrained RDNN leads to standard over-parameterization, which mathematically smoothes the loss landscape and facilitates optimization (Du et al., 2019; Cooper, 2018). Consequently, the unconstrained RDNN scales gracefully with $H$, while the factorized network suffers from the bottleneck anomaly (see Appendix E.5.4).

## 5.3 Spectral Integrity and the Degradation of Normal Hyperbolicity

Although the explicit low-rank networks achieve highly competitive test performance (Figure A9), forcing an explicit factorization has noticeable consequences on their spectral properties. As shown in the eigenvalue modulus density (Figure A11) and complex plane scattering (Figure A12), the bimodal spectrum, comprising a sharp peak near 0 (fast modes) and a peak near 1 (slow modes), is largely preserved, demonstrating the robustness of the divisor-gated architecture.

However, in the memory-guided saccade task (Figure A11, right), we observe a subtle degradation of the spectral gap in some explicit low-rank networks. Unlike the unconstrained RDNN, which exhibits a strict zero density in the middle of the spectrum, some explicit low-rank variants show a non-zero leakage of eigenvalues in the intermediate region between the two peaks.

Dynamically, this spectral leakage indicates that the hard factorization limits the network's ability to perfectly coordinate the feedback loops between the excitatory state $\mathbf{R}$ and the inhibitory state $\mathbf{G}$ (Murphy & Miller, 2009; Mastrogiuseppe & Ostojic, 2018). This restriction introduces weak, intermediate-time-scale modes that do not decay rapidly enough, slightly compromising the normal hyperbolicity of the slow manifold and, to some extent, explaining the minor performance gap compared to the unconstrained RDNN (Chaudhuri et al., 2019; Ságodi et al., 2024) (Figure A9).

## 6 Ablation Study: Divisive vs. Subtractive Inhibition

In biological neural circuits, inhibition primarily manifests in two distinct forms: subtractive inhibition, which shifts the membrane potential linearly, and divisive inhibition, which multiplicatively scales the neural response (Carandini & Heeger, 2012; Holt & Koch, 1997). To isolate the specific computational benefits of divisive normalization, we conduct an ablation study by replacing the divisive gate in the RDNN with a subtractive inhibitory term (referred to as the Subtractive Network). Specifically, the continuous-time excitatory dynamics are modified to:

$$\tau_{\mathbf{R}} d\mathbf{R} = (-\mathbf{R} + \mathbf{J}f(\mathbf{R}) + \mathbf{I}(t) - \mathbf{G}) \, dt + \sigma_{\mathbf{R}} d\mathbf{W_R}, \quad \mathbf{R} \leftarrow \max(\mathbf{0}, \mathbf{R}), \tag{2}$$

where the state is continuously rectified to enforce non-negative representations. While both networks achieve competitive task performance (Figure A13), we investigate whether the multiplicative nature of divisive gain control is strictly necessary for the underlying slow manifold dynamics.

### 6.1 Attractor Shattering in Input-Driven Integration

The main divergence between the two inhibitory mechanisms emerges in the input-driven angular velocity integration task. Notably, the Subtractive Network's slow manifold shatters into a discretized state space heavily populated by stable fixed points (Figure 4**A**), which approximate or outnumber saddle points across most configurations. This topological fragmentation indicates that the continuous manifold has broken into deep, localized point-attractor basins that easily trap the network state during integration. In contrast, the RDNN's topology is predominantly governed by saddle points. In driven dynamical systems, rather than acting as strict barriers, these saddles function as *slow channels* or ghost attractors (Strogatz, 2018), facilitating smooth, continuous state transitions along the manifold without trapping the trajectory.

While both networks exhibit similar eigenvalue modulus density, with a prominent eigenvalue modulus peak near $|\lambda| \approx 1$, to satisfy the BPTT memory constraint during integration (Figure A14, left), their underlying dimensionalities differ significantly. As quantitatively corroborated by the elevated effective rank of the Subtractive Network (Figure 5 and Table A1), the ablated network recruited additional physical dimensions to support its dynamics. As derived in Appendix E.5.1, this dimensional compensation stems from a gradient-level decoupling: unlike divisive normalization, additive inhibition fails to multiplicatively scale the BPTT gradients. While the hard-thresholding of ReLU still suppresses strictly silent neurons, the subtractive network lacks the implicit regularization to dynamically contract the active normal dimensions ($|\lambda_{\text{normal}}| \to 0$). Without this capacity to intrinsically confine its dynamics to a strict one-dimensional ring, the optimizer is forced to construct redundant slow pathways across a fragmented, higher-dimensional landscape to bridge the shattered discrete basins and prevent memory decay under time-varying inputs.

### 6.2 Autonomous Maintenance and Effective Rank Compression

Conversely, in the autonomous memory-guided saccade task, where external input is absent during the delay period, the Subtractive Network successfully forms marginal fixed points and maintains a strict spectral gap, closely mirroring the RDNN (Figure 4**B** and A14, right). Indeed, both networks exhibit nearly identical

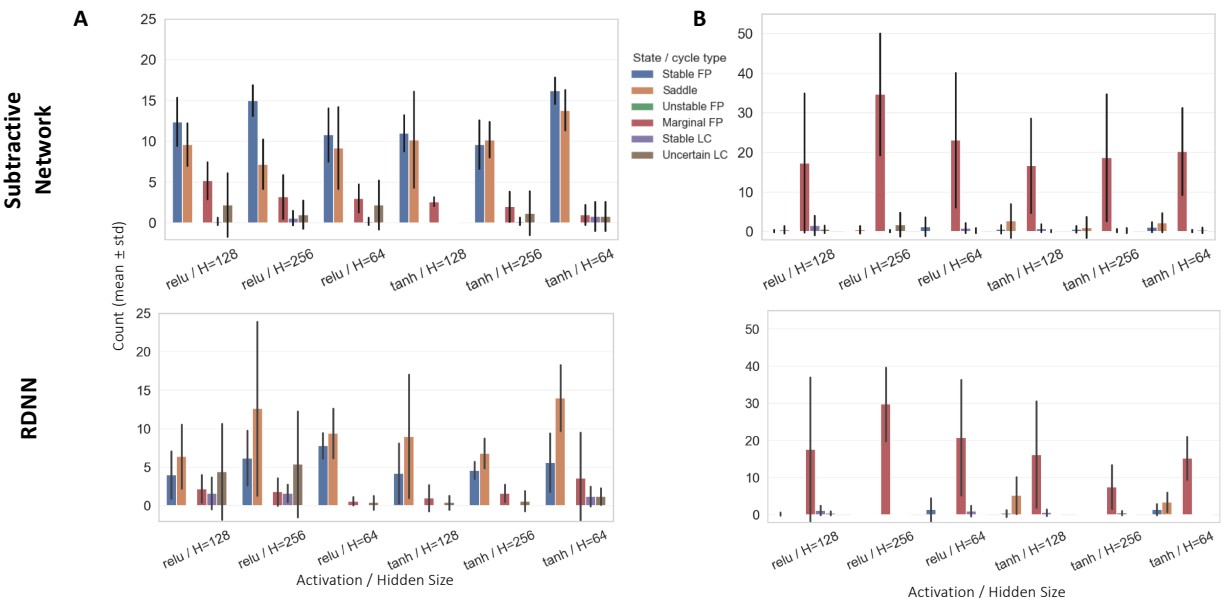

Figure 4: Comparison of dynamical topology between divisive and subtractive networks. Statistical counts of topological types across random seeds and configurations. **A** Angular Velocity Integration. Under continuous velocity inputs, the Subtractive Network's manifold fragments into a portfolio heavily populated by stable fixed points (blue). In contrast, the RDNN exhibits a more fluid topology governed by saddles (acting as slow channels). **B** Memory-guided Saccade. In this autonomous task ($\mathbf{I}(t) = \mathbf{0}$), both networks successfully sustain marginal fixed points (red), demonstrating that subtractive inhibition is sufficient for static memory maintenance when dynamic gain control is not required.

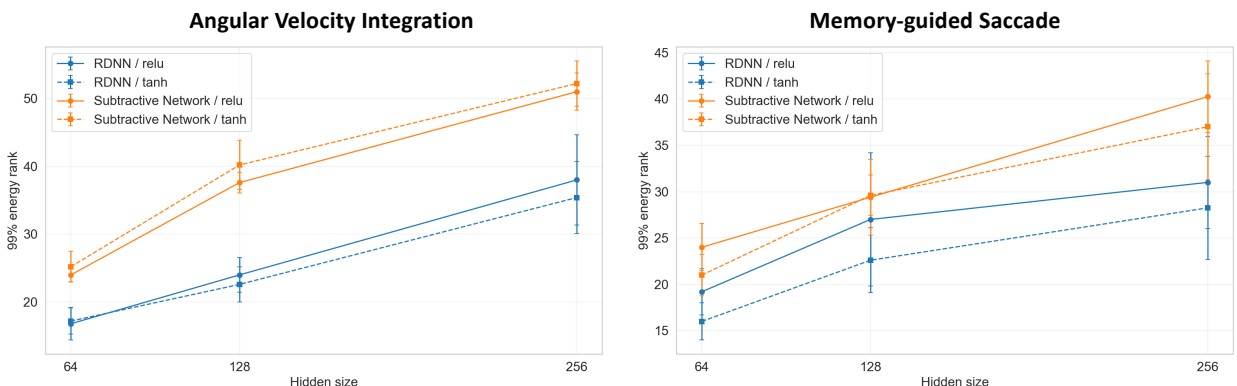

Figure 5: The 99% energy effective rank of the merged recurrent weight matrices is plotted against hidden sizes ($H \in \{64, 128, 256\}$). While both the RDNN (blue) and the Subtractive Network (orange) maintain a sub-linear scaling with $H$, the RDNN's effective rank is consistently and significantly lower across both tasks. This quantitative gap demonstrates that multiplicative divisive normalization acts as a stronger implicit low-rank regularizer during training than additive subtractive inhibition, compressing the recurrent dynamics into a tighter, lower-dimensional subspace.

fixed-point portfolios (dominated by marginal fixed points) and eigenvalue modulus density spectra. This similarity in the autonomous regime ($\mathbf{I}(t) = 0$) is mathematically expected: under zero-input conditions, the primary computational requirement is to balance recurrent excitation with inhibitory feedback to achieve marginal stability ($\lambda_{\max} \approx 1$). Since the input is absent during the delay, the unique scale-invariance and

gain-control properties of divisive normalization are not put to the test, allowing both linear subtractive and non-linear divisive inhibition to sustain a functionally equivalent ring attractor (Ben-Yishai et al., 1995; Zhang, 1996; Ayaz & Chance, 2009). As derived in Appendix E.5.1, while time-varying inputs cause discontinuous state-switching in the Heaviside gating that shatters the ring (Hahnloser et al., 2000), this gating remains static under zero-input conditions, preserving the manifold.

However, quantitative analysis of the 99% energy effective rank (Figure 5 and Table A1) reveals a fundamental difference in their regularizing properties. While the Subtractive Network successfully avoids the linear rank explosion seen in standard gated RNNs (exhibiting sub-linear scaling with hidden size $H$), its effective rank remains consistently higher than that of the RDNN across all configurations. This indicates that divisive normalization acts as a significantly stronger implicit regularizer. The multiplicative scaling dynamically compresses the variance of the state space, forcing the recurrent dynamics into a tighter, lower-dimensional subspace than additive subtraction can achieve.

### 6.3 Functional Implications in Neuroscience

These findings provide a compelling computational perspective on the coexistence of divisive and subtractive inhibition in biological systems, delineating their distinct functional niches. Subtractive inhibition is highly suited for static thresholding, baseline noise filtering, and sparse coding (Carandini & Heeger, 2012; Silver, 2010). By shifting the neural activation curve to the right, subtractive mechanisms act as an effective noise gate that filters out weak spontaneous background activity and sharpens sensory tuning curves. This makes subtractive inhibition computationally efficient for static memory maintenance in autonomous regimes (as demonstrated in our memory-guided saccade task) and binary state switching. However, when a circuit must continuously integrate time-varying external inputs, additive subtractive mechanisms fail to scale dynamically with the input magnitude, leading to attractor shattering. Divisive normalization, by providing contrast invariance and dynamic gain control (Heeger, 1992), preserves the structural integrity of the low-dimensional manifold under external drive. This suggests that divisive normalization is not merely a biological artifact, but a computational necessity for high-fidelity, input-driven continuous working memory.

## 7 Discussion and Conclusion

**Divisive Normalization as a Canonical Computation**  Our findings provide a novel dynamical systems perspective on why divisive normalization (DN) is widely regarded as a canonical neural computation (Carandini & Heeger, 2012). While traditionally understood as a mechanism for sensory gain control and contrast invariance, our results demonstrate its role in shaping the topological landscape of recurrent memory circuits. By dynamically compressing the state variance and implicitly regularizing the effective rank, DN allows neural populations to robustly maintain continuous attractors under time-varying inputs, a computational feat where subtractive inhibition structurally fails. This suggests that DN is computationally essential not just for sensory encoding, but as a fundamental building block for the stable maintenance and manipulation of continuous cognitive variables.

**Implicit Regularization and Optimization Dynamics**  Our low-rank analysis sheds light on how different dimensionality reduction strategies influence the optimization landscape of recurrent networks. While the unconstrained RDNN naturally self-compresses its effective rank via the activity-dependent divisive bottleneck, explicitly factorizing the recurrent weights introduces severe optimization challenges. Specifically, we observed a higher-rank convergence anomaly where wider explicit bottlenecks paradoxically hinder training. This aligns with non-convex matrix factorization theory, which shows that over-parameterized product spaces introduce redundant scaling symmetries and degenerate saddle points (Li et al., 2019). Conversely, the unconstrained network avoids these non-convexities and benefits from standard over-parameterization, which mathematically smoothes the loss landscape (Du et al., 2019). These findings suggest that biologically inspired multiplicative gain control offers a highly optimization-friendly route to discovering low-dimensional cognitive representations without relying on hard architectural bottlenecks.

**Efficient Representations of the Continuous World**   For machine learning, our study offers insights into building efficient architectures for representing the continuous world. Standard gated RNNs (e.g., GRUs, LSTMs) tend to discretize continuous manifolds into high-dimensional, fragmented point attractors. In contrast, the RDNN demonstrates that multiplicative divisive feedback acts as a powerful structural inductive bias. It enables the network to learn high-fidelity, low-rank continuous representations without the optimization pathologies associated with explicit low-rank factorization. This highlights a promising pathway for designing parameter-efficient, continuous-time models for tasks requiring smooth temporal integration, such as spatial navigation and robotics.

**Limitations and Future Directions**   Despite these theoretical insights, our study has several limitations that pave the way for future research. First, while Backpropagation Through Time (BPTT) provides a mathematically rigorous framework to understand the emergence of low-rank dynamics and spectral compression, it lacks biological plausibility. Future work should investigate whether more physiological synaptic plasticity rules, such as Hebbian learning or predictive coding approximations (Lillicrap et al., 2020), can similarly induce these topological properties in DN-equipped networks. Second, the current model focuses exclusively on continuous working memory. Given that biological systems seamlessly integrate both discrete and continuous information (Constantinidis et al., 2018), exploring how divisive normalization might support mixed discrete-continuous memory representations is a compelling next step. Finally, as a theoretical and computational model, our predictions regarding the eigenvalue spectra and effective rank of recurrent circuits require further empirical validation using large-scale neural population recordings from behaving animals (Chaudhuri et al., 2019).

**Conclusion**   In conclusion, we introduced the RDNN to bridge the gap between biological gain control and artificial recurrent memory. Through dynamical and optimization analyses, we established that divisive normalization naturally induces low-rank, normally hyperbolic slow manifolds. By overcoming the attractor shattering and fine-tuning problems inherent in standard architectures, our work provides a unified theoretical framework for understanding and modeling robust continuous working memory in both biological and artificial systems.

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

## A Tasks

Two canonical continuous working memory tasks are employed to evaluate the capacity of recurrent networks to form and maintain continuous working memory, and both tasks require the network to encode a circular variable $\theta \in [0, 2\pi)$ and output its Cartesian coordinates $\mathbf{y}_t = [\sin(\theta_t), \cos(\theta_t)]$.

**Angular Velocity Integration** This task tests the network's ability to continuously update its internal representation based on a dynamic input. The total length of a trial is 256 time steps. The input is a one-dimensional continuous angular velocity signal $v(t)$ generated by superimposing three sinusoidal waves. For each wave, the frequency, phase, and amplitude magnitude are sampled from uniform distributions $\mathcal{U}(0.2, 2.0)$ Hz, $\mathcal{U}(0, 2\pi)$, and $\mathcal{U}(0.5, 2.0)$, respectively, with a random sign. A constant bias is added to ensure the total rotation is uniformly sampled from $\mathcal{U}(1.2\pi, 4.0\pi)$. The network is provided with the initial position $[\sin(\theta_0), \cos(\theta_0)]$ ($\theta_0 \sim \mathcal{U}(0, 2\pi)$) at $t = 0$ and must output the integrated position at each subsequent step (Figure A1). An additional linear mapping is applied in this task to initialize the hidden state onto the initial position along the ring from which the network needed to integrate from.

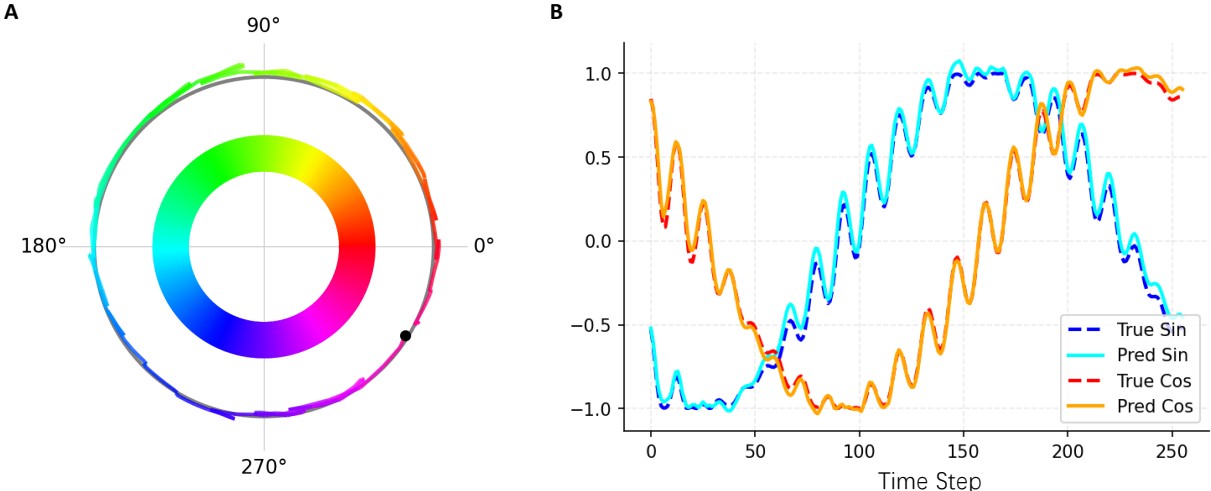

Figure A1: Visualization of the angular velocity integration task. **A** The 2D output trajectory from RDNN, with the color gradient indicating the instantaneous encoded angle. **B** Time-course of the true and predicted $[\sin(\theta), \cos(\theta)]$ values for a single trial performed by RDNN.

**Memory-guided Saccade** This task evaluates the network's ability to maintain a static memory over a prolonged, variable delay period without external input. The total length of a trial is 512 time steps. The input is a 3-dimensional sequence: the first two dimensions provide the target angle $[\sin(\theta), \cos(\theta)]$ ($\theta \sim \mathcal{U}(0, 2\pi)$) for a brief stimulus period (15 steps), followed by a variable zero-input delay period sampled from a discrete uniform distribution $\mathcal{U}\{50, 400\}$, and the fixation signal as the third dimension. A Go Cue is then presented in the third input dimension for 5 steps. Crucially, the network is trained to output $[0, 0]$ (fixation) during the stimulus and delay periods, and must rapidly decode the remembered angle to the output layer only after receiving the Go Cue, with the loss masked out during the 5-step cue transition (Figure A2).

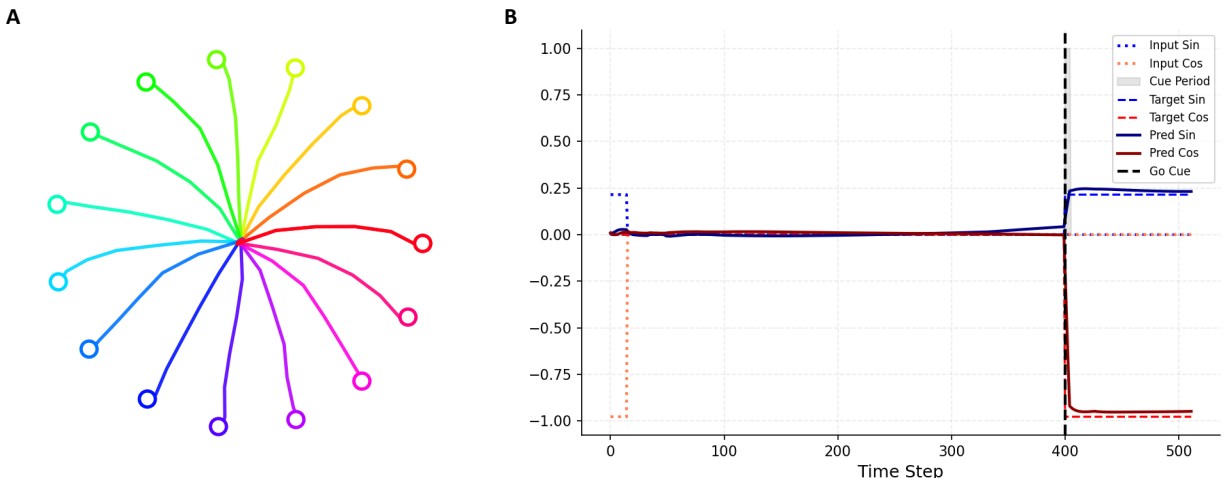

Figure A2: Visualization of the memory-guided saccade task. **A** RDNN's 2D output trajectories across multiple target angles. The network maintains a zero-output fixation state at the origin before executing rapid radial saccades to the target angles. **B** Time-course of a single trial.

## B    Discretization and Implementation of RDNN Dynamics

To train the RDNN using backpropagation through time (BPTT) in standard deep learning frameworks, we discretize the continuous-time stochastic differential equations (Eq. 1) using the Euler-Maruyama method.

**Input Projection and Output Decoding**   To interface the recurrent core with external task variables, the raw input sequence $\mathbf{x}_t \in \mathbb{R}^{D_{in}}$ at each time step is mapped to the network's internal input current $\mathbf{I}_t \in \mathbb{R}^H$ via a linear projection layer $\mathbf{I}_t = \mathbf{W}_{in}\mathbf{x}_t + \mathbf{b}_{in}$, where $\mathbf{W}_{in} \in \mathbb{R}^{H \times D_{in}}$ and $\mathbf{b}_{in} \in \mathbb{R}^H$ are trainable weights and biases.

Similarly, the task-specific predictions $\mathbf{y}_t \in \mathbb{R}^{D_{out}}$ are decoded linearly from the principal excitatory state $\mathbf{R}_t$: $\mathbf{y}_t = \mathbf{W}_{out}\mathbf{R}_t + \mathbf{b}_{out}$. For the continuous working memory tasks evaluated in this work, the output dimension is $D_{out} = 2$, corresponding to the Cartesian coordinates $[\sin(\theta_t), \cos(\theta_t)]$ of the encoded circular variable.

**Discrete-Time Recurrent Updates**   Let $\Delta t$ be the integration time step. We define the discrete update rates (or leak factors) as $\boldsymbol{\alpha}_R = \frac{\Delta t}{\tau_R}$ and $\boldsymbol{\alpha}_G = \frac{\Delta t}{\tau_G}$. The discrete-time iterative updates for the hidden states $\mathbf{R}_t$ and $\mathbf{G}_t$ at step $t$ are formulated as:

$$
\begin{aligned}
\tilde{\mathbf{R}}_t &= \frac{\mathbf{J}f(\mathbf{R}_{t-1}) + \mathbf{I}_t}{\boldsymbol{\eta} + \mathbf{G}_{t-1}} + \boldsymbol{\xi}_{R,t}, \\
\mathbf{R}_t &= (1 - \boldsymbol{\alpha}_R) \odot \mathbf{R}_{t-1} + \boldsymbol{\alpha}_R \odot \tilde{\mathbf{R}}_t, \\
\tilde{\mathbf{G}}_t &= \mathbf{W}f(\mathbf{R}_{t-1}) + \boldsymbol{\xi}_{G,t}, \\
\mathbf{G}_t &= (1 - \boldsymbol{\alpha}_G) \odot \mathbf{G}_{t-1} + \boldsymbol{\alpha}_G \odot \tilde{\mathbf{G}}_t,
\end{aligned}
\tag{A-1}
$$

where $\odot$ denotes element-wise multiplication, and $\boldsymbol{\xi}_{R,t}, \boldsymbol{\xi}_{G,t} \sim \mathcal{N}(0, \sigma^2\mathbf{I})$ are the injected Gaussian state noises at each time step.

**Parameterization and Biological Constraints**   To ensure the system remains a positive dynamical system (consistent with Dale's principle and the biological reality of excitatory and inhibitory identities), we enforce strict positivity on specific parameters during training. In our implementation, this is achieved by defining unconstrained raw parameters and applying the softplus function during the forward pass:

$$
\mathbf{J} = \text{Softplus}(\mathbf{J}_{\text{raw}}), \quad \mathbf{W} = \text{Softplus}(\mathbf{W}_{\text{raw}}), \quad \boldsymbol{\eta} = \text{Softplus}(\boldsymbol{\eta}_{\text{raw}}) + \epsilon,
\tag{A-2}
$$

where $\epsilon = 10^{-5}$ is a small constant for numerical stability.

Similarly, to ensure that the update rates $\boldsymbol{\alpha}_R$ and $\boldsymbol{\alpha}_G$ strictly bound the leaky integration between 0 and 1, we parameterize them using the sigmoid function:

$$
\boldsymbol{\alpha}_R = \sigma(\boldsymbol{\alpha}_{R,\text{raw}}), \quad \boldsymbol{\alpha}_G = \sigma(\boldsymbol{\alpha}_{G,\text{raw}}).
\tag{A-3}
$$

**Initialization Strategy**   Proper initialization is crucial for training recurrent networks with positive feedback loops (Le et al., 2015). To prevent the product of positive matrices from causing numerical explosion in early training phases, we initialize $\mathbf{J}_{\text{raw}}$ and $\mathbf{W}_{\text{raw}}$ from a normal distribution $\mathcal{N}(-\ln(H), 1/\sqrt{H})$. This ensures that the initial spectral radius of the recurrent drive is well-behaved. Furthermore, $\boldsymbol{\alpha}_{R,\text{raw}}$ and $\boldsymbol{\alpha}_{G,\text{raw}}$ are initialized with a negative mean (e.g., $-2.2$), yielding an initial update rate of $\alpha \approx 0.1$. This explicitly biases the network towards functioning as a slow integrator at the beginning of training, which is essential for learning long-term dependencies in continuous working memory tasks.

## C    Training Details

**General Training Setup**   All models were trained using the Adam optimizer ($\beta_1 = 0.9$, $\beta_2 = 0.999$) with a batch size of 64 for a total of 100 epochs with training data generated online. During training, we

applied gradient clipping with a maximum norm of 1.0 to prevent gradient explosion. We also injected Gaussian state noise ($\sigma = 0.1$) into the hidden dynamics at each time step during the training phase. While training low-rank RDNN, we extended the training epochs to 200 to ensure convergence, as the reduced parameterization can lead to slower learning dynamics.

For the angular velocity integration task, the models were optimized using the standard Mean Squared Error (MSE) over the entire sequence of length $T$:

$$\mathcal{L}_{int} = \frac{1}{T} \sum_{t=1}^{T} \|\mathbf{y}_t - \hat{\mathbf{y}}_t\|_2^2, \tag{A-4}$$

where $\mathbf{y}_t = [\sin(\theta_t), \cos(\theta_t)]$ is the target coordinate and $\hat{\mathbf{y}}_t$ is the network's prediction at time $t$.

For the memory-guided saccade task, we utilized a masked MSE loss defined as:

$$\mathcal{L}_{sac} = \frac{\sum_{t=1}^{T} m_t \|\mathbf{y}_t - \hat{\mathbf{y}}_t\|_2^2}{D \sum_{t=1}^{T} m_t}, \tag{A-5}$$

where $m_t \in \{0, 1\}$ is a binary mask ($m_t = 0$ during the Go Cue period, and $m_t = 1$ otherwise), and $D = 2$ is the output dimension. This formulation ensures that the gradient is normalized correctly by the exact number of active output elements.

**Hyperparameter Selection and Model Configurations**  To ensure a fair comparison, we conducted a pilot hyperparameter search for the learning rate. For each combination of network architecture, hidden size ($H \in \{64, 128, 256\}$), and activation function, we evaluated learning rates from the set $\{10^{-2}, 5 \times 10^{-3}, 10^{-3}\}$. Each configuration was trained for 10 epochs, and the learning rate $5 \times 10^{-3}$ that yielded the lowest training loss was selected for the full 100-epoch training. All reported results and dynamical analyses are based on 5 independent runs with different random seeds for each configuration. Training a single network took about $10 \sim 20$ minutes on CPU, and $\sim 800$MB of memory.

**Baseline Initialization**  For the baseline GRU and LSTM models, we employed specific initialization strategies to improve their ability to capture long-term dependencies. The hidden-to-hidden projection weights were initialized from a normal distribution $\mathcal{N}(0, \sigma^2)$ with $\sigma = 1/\sqrt{H}$. Furthermore, following standard practices for training gated RNNs on memory tasks, the biases corresponding to the update gate in the GRU and the forget gate in the LSTM were explicitly initialized to 1.0. This biases the gates towards retaining past information during the early stages of training.

**Explicit Low-Rank Initialization**  For the explicitly factorized Low-Rank RDNN, proper initialization is critical to prevent numerical instability caused by the product of positive matrices. We initialize the raw factor matrices $(\mathbf{J}_1, \mathbf{J}_2, \mathbf{W}_1, \mathbf{W}_2)$ from a normal distribution with a negative mean of $-1.0$ and standard deviations scaled by $1/\sqrt{r}$ and $1/\sqrt{H}$, respectively. Since these parameters are passed through a Softplus function during the forward pass, the negative mean ensures that the initial effective weights are sufficiently small (Softplus$(-1.0) \approx 0.31$). This prevents the explosive positive feedback that would otherwise occur when multiplying two strictly positive matrices in the early stages of training. Furthermore, the raw update rates $\boldsymbol{\alpha}_{R,\mathrm{raw}}$ and $\boldsymbol{\alpha}_{G,\mathrm{raw}}$ are initialized with a mean of $-2.2$. After applying the sigmoid function, this yields an initial update rate of $\alpha \approx 0.1$, and explicitly biases the network to operate as a slow integrator at the onset of training, facilitating the learning of long-term dependencies in continuous working memory tasks.

**Exceptions and Convergence Issues**  During our experiments, we observed two notable training exceptions:

- **LSTM with ReLU:** We excluded the LSTM architecture paired with the ReLU activation function from our analysis. Empirical observations indicated that this specific combination was highly susceptible to gradient explosion and remained largely untrainable across all tested learning rates (Le et al., 2015; Greff et al., 2017; Jozefowicz et al., 2015).

- **Baseline Failures on the Memory-guided Saccade Task:** While the proposed RDNN successfully and stably converged on the memory-guided saccade task, both the GRU and LSTM baselines completely failed to learn the task across all tested learning rates. As illustrated by the training loss curves in Figure A3, the RDNN demonstrates a steady decrease in loss, whereas the GRU and LSTM remain stuck at high error levels. From a dynamical systems perspective, maintaining an arbitrary continuous variable over a prolonged zero-input delay requires synthesizing a perfectly flat energy landscape (i.e., a continuous attractor). Standard gated RNNs are structurally biased toward discrete point attractors; consequently, their hidden states inevitably drift into localized basins during the long delay, resulting in large, uncorrectable readout errors (Jordan et al., 2021). While they might converge with task-specific tricks, the RDNN naturally solves this via its divisive gain control. Consequently, the analysis for the memory-guided saccade task was exclusively performed on the RDNN configurations.

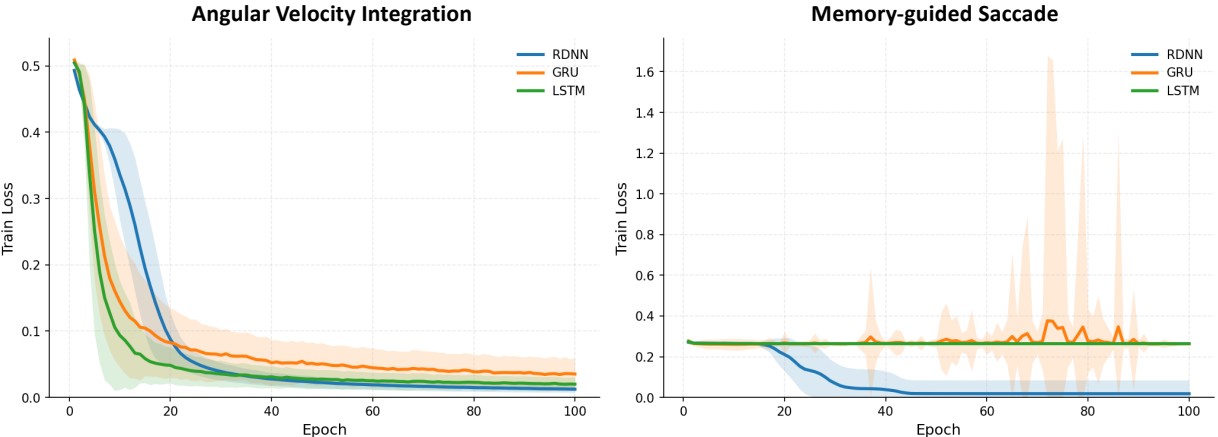

Figure A3: Training loss curves across epochs. Error bands indicate standard deviations across different hidden sizes, activation functions, and random seeds.

# D    Analysis Methods

## D.1    Evaluation Metric

On trained networks, we report the normalized mean squared error (NMSE) of the network prediction $\hat{\mathbf{y}}$ compared to the ground truth target $\mathbf{y}$ across 1024 test trials with state noise:

$$\text{NMSE} = \frac{\mathbb{E}\left[(\hat{\mathbf{y}} - \mathbf{y})^2\right]}{\mathbb{E}\left[\mathbf{y}\right]}. \tag{A-6}$$

This metric quantifies the proportion of variance in the target that is not captured by the network's predictions, with lower values indicating better performance.

## D.2    Dynamical System Analysis

To rigorously characterize the underlying computational mechanisms of the trained networks, we analyzed their autonomous dynamics (i.e., network evolution without external inputs, $\mathbf{I}_t = \mathbf{0}$) initialized from task-relevant states. The analysis pipeline extracts the slow manifold, identifies topological structures (fixed points and limit cycles), and computes the spectral properties of the system.

**Identification of the Slow Manifold**   To identify the slow manifold (approximate continuous attractor) embedded in the high-dimensional state space, we simulated the autonomous dynamics for an extended period ($32\times$ task length) of 1024 trajectories without state noise starting from states collected at the end of task trials. We computed the instantaneous drift speed in the output space as $v_t = \|\mathbf{y}_{t+1} - \mathbf{y}_t\|_2$. Points with drift speeds below a progressive relaxation threshold (starting from $10^{-3}$ of the maximum trajectory speed) were selected as candidate slow points. To ensure a uniform representation of the ring manifold, we discretized the angular output space $[-\pi, \pi)$ into 256 equal bins. Within each angular bin, the state exhibiting the minimum drift speed was selected. This procedure yields an ordered set of states that empirically traces the one-dimensional slow manifold.

**Fixed Point Detection and Topological Classification**   Fixed points were initially located by analyzing the one-dimensional angular flow on the identified slow manifold. We computed the angular displacement $\Delta\theta = \theta_{t+1} - \theta_t$ for each state. Candidate fixed points were identified at zero-crossings where the sign of $\Delta\theta$ flipped between adjacent states.

To classify the topology of these candidates, we computed the exact Jacobian matrix $\mathcal{J} = \frac{\partial F(\mathbf{h})}{\partial \mathbf{h}}$ at each point using exact automatic differentiation, where $F(\cdot)$ denotes the discrete-time recurrent update function. Let $|\lambda|_{\max}$ and $|\lambda|_{\min}$ be the maximum and minimum moduli of the eigenvalues of $\mathcal{J}$, respectively. Given a small tolerance $\epsilon = 10^{-3}$, the fixed points were classified as follows:

- **Stable Fixed Point:** $|\lambda|_{\max} < 1 - \epsilon$.

- **Unstable Fixed Point:** $|\lambda|_{\min} > 1 + \epsilon$.

- **Saddle Point:** The spectrum contains both $|\lambda| > 1 + \epsilon$ and $|\lambda| < 1 - \epsilon$.

- **Marginal Fixed Point:** $|\lambda|_{\max} \approx 1$ (within $[1 - \epsilon, 1 + \epsilon]$), indicating neutral stability along the manifold, a hallmark of perfect continuous attractors.

**Limit Cycle Detection and Floquet Analysis**   For tasks involving continuous integration (e.g., angular velocity), the network may form stable limit cycles rather than fixed points. We detected limit cycles by analyzing the tail of the autonomous trajectories. We searched for periodic orbits by comparing adjacent trajectory chunks of varying periods $p$. A cycle of period $p$ was identified if the relative $L_2$ distance between consecutive chunks fell below a strict tolerance ($5 \times 10^{-3}$) and the oscillation amplitude was non-trivial.

To assess the stability of the detected limit cycles, we performed Floquet analysis. We computed the Monodromy matrix $\mathbf{M}$ as the product of the Jacobians along the periodic orbit: $\mathbf{M} = \prod_{k=1}^{p} \mathcal{J}_k$. The limit cycle is classified as **stable** if the maximum modulus of the eigenvalues of $\mathbf{M}$ (Floquet multipliers) is strictly less than 1, and **uncertain** otherwise.

**Eigenvalue Spectrum**   To evaluate the time-scale separation and normal hyperbolicity of the dynamics, we extracted the full complex spectrum of the Jacobians. For all identified slow points on the manifold, we calculated the eigenvalues of the Jacobian matrix $\mathcal{J}$, analyzed the kernel density estimation of their moduli and derived the spectrum on the complex plane.

**Uniform Norm of the Manifold Flow**   To theoretically bound the short-term memory error (Ságodi et al., 2024), we calculated the uniform norm ($L_\infty$ norm) of the vector field restricted to the slow manifold. Specifically, for all states $\mathbf{h} \in \mathcal{M}$ on the identified manifold, we computed the maximum 1-step drift magnitude in the output space:

$$\|\Delta\mathbf{y}\|_\infty = \max_{\mathbf{h}\in\mathcal{M}} \|\mathbf{W}_{out}F(\mathbf{h}) - \mathbf{W}_{out}\mathbf{h}\|_2. \tag{A-7}$$

**Manifold Reliability Assessment**   To automatically filter out degenerate solutions or networks that failed to form a coherent ring topology, we implemented a reliability metric. A manifold was deemed **reliable** if it successfully covered at least 25% of the angular bins (coverage $\geq 0.25$) and contained a minimum absolute

number of valid slow points (e.g., $\geq \max(16, N_{bins}/8)$). Networks failing this criterion were considered to have collapsed into trivial point attractors. Figure A4 summarizes the reliability of the learned manifolds across different architectures.

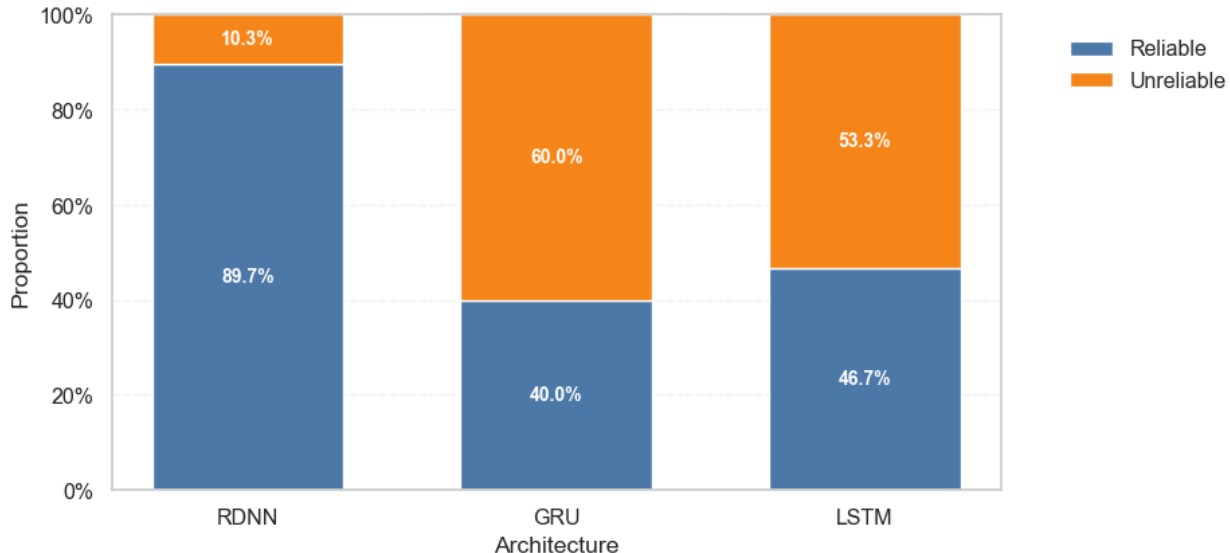

Figure A4: Reliability of the learned slow manifolds across architectures. A manifold is classified as reliable if it successfully forms a coherent ring topology covering a sufficient proportion of the angular space.

### D.3 Effective Rank of Recurrent Weight Matrices

To quantify the intrinsic dimensionality of the learned recurrent dynamics and evaluate the structural inductive biases of different architectures, we computed the effective rank of their recurrent weight matrices through constructing a unified recurrent weight matrix $\mathbf{W}_{\mathrm{merged}}$ for each model by concatenating its constituent hidden-to-hidden projection matrices.

Specifically, for the proposed RDNN, the merged matrix was formed by concatenating the effective excitatory and inhibitory recurrent weights after applying the positivity constraints: $\mathbf{W}_{\mathrm{merged}} = [\mathrm{Softplus}(\mathbf{J}_{\mathrm{raw}}) \,\|\, \mathrm{Softplus}(\mathbf{W}_{\mathrm{raw}})]$. For the GRU baseline, $\mathbf{W}_{\mathrm{merged}}$ consisted of the concatenated hidden-state weights for the reset, update, and new gates. Similarly, for the LSTM, it comprised the hidden-state weights for the input, forget, candidate, and output gates.

We performed Singular Value Decomposition (SVD) on $\mathbf{W}_{\mathrm{merged}}$ to extract its singular values $\sigma_1 \geq \sigma_2 \geq \cdots \geq \sigma_n \geq 0$. To robustly measure the dimensionality of the subspace that dominates the recurrent computations, we defined the effective rank as the 99% energy rank. This is calculated as the minimum number of singular values $k$ required to capture 99% of the total spectral energy (i.e., the sum of squared singular values):

$$k = \min \left\{ r \;\middle|\; \frac{\sum_{i=1}^{r} \sigma_i^2}{\sum_{j=1}^{n} \sigma_j^2} \geq 0.99 \right\}. \tag{A-8}$$

## E   Training and Optimization Dynamics under BPTT

In this section, we present the brief mathematical formulations of the Recurrent Divisive Normalization Network (RDNN), its subtractive variant, and its explicitly factorized low-rank counterpart. We derive their gradient dynamics under Backpropagation Through Time (BPTT) and provide a theoretical treatment of the emergent low-rank connectivity and optimization landscapes observed in our experiments.

### E.1 Discrete-Time State Formulations

Let $\mathbf{R}_t \in \mathbb{R}_+^H$ and $\mathbf{G}_t \in \mathbb{R}_+^H$ be the excitatory and inhibitory state vectors at time step $t$. We denote the element-wise division by $\oslash$ and the element-wise product by $\odot$. The activation function is denoted by $f(\cdot)$, and $\mathbf{I}_t \in \mathbb{R}^H$ is the input current.

The discrete-time update rule of the RDNN is formulated as:

$$
\begin{aligned}
\mathbf{R}_t &= (\mathbf{1} - \boldsymbol{\alpha}_R) \odot \mathbf{R}_{t-1} + \boldsymbol{\alpha}_R \odot \left[ (\mathbf{J} f(\mathbf{R}_{t-1}) + \mathbf{I}_t) \oslash (\boldsymbol{\eta} + \mathbf{G}_{t-1}) \right], \\
\mathbf{G}_t &= (\mathbf{1} - \boldsymbol{\alpha}_G) \odot \mathbf{G}_{t-1} + \boldsymbol{\alpha}_G \odot \left[ \mathbf{W} f(\mathbf{R}_{t-1}) \right].
\end{aligned}
\tag{A-9}
$$

For the Subtractive Network, the divisive gate is replaced by an additive inhibitory subtraction, and a rectification threshold is applied to the updated state to enforce non-negative firing rates:

$$
\mathbf{R}_t = \max \left( \mathbf{0}, \, (\mathbf{1} - \boldsymbol{\alpha}_R) \odot \mathbf{R}_{t-1} + \boldsymbol{\alpha}_R \odot (\mathbf{J} f(\mathbf{R}_{t-1}) + \mathbf{I}_t - \mathbf{G}_{t-1}) \right).
\tag{A-10}
$$

For the Low-Rank RDNN with explicit rank $r < H$, the recurrent weight matrices $\mathbf{J}$ and $\mathbf{W}$ are explicitly factorized into product forms:

$$
\mathbf{J} = \mathbf{J}_1 \mathbf{J}_2, \quad \mathbf{W} = \mathbf{W}_1 \mathbf{W}_2.
\tag{A-11}
$$

where $\mathbf{J}_1, \mathbf{W}_1 \in \mathbb{R}^{H \times r}$ and $\mathbf{J}_2, \mathbf{W}_2 \in \mathbb{R}^{r \times H}$.

### E.2 Gradient Dynamics and BPTT Modulation

Let $\mathcal{L} = \sum_{t=1}^{T} \mathcal{L}_t(\mathbf{R}_t)$ be the total scalar loss over the sequence. Under BPTT, the gradient of the loss with respect to the recurrent excitatory matrix $\mathbf{J}$ is given by:

$$
\frac{\partial \mathcal{L}}{\partial \mathbf{J}} = \sum_{t=1}^{T} \sum_{i=1}^{H} \frac{\partial \mathcal{L}}{\partial R_{t,i}} \frac{\partial R_{t,i}}{\partial \mathbf{J}}.
\tag{A-12}
$$

To isolate the direct influence of the divisive normalization at step $t$, we compute the partial derivative of $R_{t,i}$ with respect to $J_{jk}$ (where $j, k \in \{1, \ldots, H\}$):

$$
\frac{\partial R_{t,i}}{\partial J_{jk}} = (1 - \alpha_{R,i}) \frac{\partial R_{t-1,i}}{\partial J_{jk}} + \alpha_{R,i} \left[ \frac{\delta_{ij} f(R_{t-1,k})}{\eta_i + G_{t-1,i}} - \frac{J_{i,\cdot} f(\mathbf{R}_{t-1}) + I_{t,i}}{(\eta_i + G_{t-1,i})^2} \frac{\partial G_{t-1,i}}{\partial J_{jk}} \right],
\tag{A-13}
$$

where $\delta_{ij}$ is the Kronecker delta.

This formulation reveals a fundamental mathematical property of the RDNN: both the direct feedforward gradient update and the indirect recurrent feedback update are scaled inversely by the dynamic divisor. Specifically, the direct gradient contribution at step $t$ is scaled by:

$$
\frac{\partial \mathcal{L}_t}{\partial J_{jk}} \propto \frac{\alpha_{R,j}}{\eta_j + G_{t-1,j}}.
\tag{A-14}
$$

### E.3 Gradient-Driven Time-Scale Separation and Normal Hyperbolicity

We now mathematically analyze how the optimization of RDNN under noisy working memory tasks naturally shapes the bimodal eigenvalue modulus spectrum ($|\lambda| \approx 1$ and $|\lambda| \approx 0$) and normal hyperbolicity observed in Figures 2.

Let $\mathcal{J}_R(k) = \frac{\partial \mathbf{R}_k}{\partial \mathbf{R}_{k-1}}$ be the localized Jacobian of the excitatory state update at step $k$, which can be expressed from the tangent linear flow of the update equation (ignoring indirect cross-coupling through $\mathbf{G}$ for clarity of the primary recurrent flow) as:

$$
\mathcal{J}_R(k) = \operatorname{diag}(\mathbf{1} - \boldsymbol{\alpha}_R) + \operatorname{diag}(\boldsymbol{\alpha}_R) \operatorname{diag}\left( \frac{\mathbf{1}}{\boldsymbol{\eta} + \mathbf{G}_{k-1}} \right) \mathbf{J} \operatorname{diag}\left( f'(\mathbf{R}_{k-1}) \right).
\tag{A-15}
$$

The gradient of the loss $\mathcal{L}_T$ at the end of the trial with respect to the hidden state at an earlier step $t$ is propagated backward through time via the chain product of these Jacobians:

$$\frac{\partial \mathcal{L}_T}{\partial \mathbf{R}_t} = \frac{\partial \mathcal{L}_T}{\partial \mathbf{R}_T} \prod_{k=t+1}^{T} \mathcal{J}_R(k). \tag{A-16}$$

**Convergence of Tangent Modes ($|\lambda_{\textbf{tangent}}| \approx 1$)**  For continuous working memory tasks where information must be preserved over a long horizon $T - t \gg 1$, the loss function $\mathcal{L}_T$ penalizes representation drift along the task-relevant manifold $\mathcal{M}$.

- If the spectral radius $\rho$ of the Jacobian product along the tangent direction $\mathbf{v}_{\text{tangent}}$ of $\mathcal{M}$ satisfies $\rho\left(\prod \mathcal{J}_R(k)\right) < 1$, the gradient vanishes exponentially ($\frac{\partial \mathcal{L}_T}{\partial \mathbf{R}_t} \to \mathbf{0}$), and the network fails to learn long-term dependencies.

- If $\rho > 1$, the gradient explodes, disrupting training stability.

Consequently, gradient-based optimization mathematically drives the parameters toward a stable critical regime along the manifold, forcing the tangent modes to satisfy:

$$\mathcal{J}_R(k)\mathbf{v}_{\text{tangent}} \approx \lambda_{\text{tangent}}\mathbf{v}_{\text{tangent}}, \quad \text{with } |\lambda_{\text{tangent}}| \approx 1. \tag{A-17}$$

This explains the sharp, robust peak in the eigenvalue modulus density near 1.0 (Figures 2).

**Convergence of Normal Modes ($|\lambda_{\textbf{normal}}| \to 0$)**  During training, we inject Gaussian state noise $\boldsymbol{\xi}_t \sim \mathcal{N}(\mathbf{0}, \sigma^2 \mathbf{I})$ into the hidden dynamics. In the linear approximation, the forward propagation of the off-manifold perturbation covariance $\boldsymbol{\Sigma}_t$ along the normal (orthogonal) directions $\mathbf{v}_{\text{normal}}$ to the manifold $\mathcal{M}$ is governed by the discrete-time Lyapunov-like update:

$$\boldsymbol{\Sigma}_t = \mathcal{J}_R(t)\boldsymbol{\Sigma}_{t-1}\mathcal{J}_R(t)^T + \sigma^2 \mathbf{I}. \tag{A-18}$$

To minimize the expected loss $\mathbb{E}[\mathcal{L}]$, the optimizer must suppress the cumulative propagation of this state noise off the manifold, preventing representation diffusion. This imposes a selective gradient bias that drives the eigenvalues corresponding to the normal directions toward zero:

$$\mathcal{J}_R(k)\mathbf{v}_{\text{normal}} \approx \lambda_{\text{normal}}\mathbf{v}_{\text{normal}}, \quad \text{with } |\lambda_{\text{normal}}| \to 0. \tag{A-19}$$

This leads to the massive accumulation of eigenvalues near 0 (Figures 2).

For configurations utilizing the ReLU activation, there exists $f'(R_{t-1,i}) = 0$ for any neuron $i$ below the threshold. This structurally forces the diagonal derivative matrix $\text{diag}(f'(\mathbf{R}_{t-1}))$ to be sparse, setting the corresponding columns of the Jacobian $\mathcal{J}_R(k)$ to exactly zero and contributing to the massive eigenvalue peak at 0.

**The Role of Divisive Scaling in Facilitating Spectral Separation**  While standard gating RNNs (like GRU/LSTM) struggle to perfectly separate these eigenvalues due to rigid linear coupling, the RDNN's divisive normalization provides a unique mathematical template. When a noisy perturbation $\boldsymbol{\xi}$ pushes the state off the manifold, it temporarily inflates the overall network activity. The inhibitory population instantly tracks this expansion, increasing $G_{k-1,i}$. Through the term $\text{diag}\left(\frac{\mathbf{1}}{\boldsymbol{\eta}+\mathbf{G}_{k-1}}\right)$, this dynamic increase in the divisor mathematically scales down the localized Jacobian $\mathcal{J}_R(k)$ for those off-manifold modes, dragging their eigenvalues towards 0.

Conversely, when the state is unperturbed and resides on the manifold, the divisor remains balanced, preserving $|\lambda_{\text{tangent}}| \approx 1$. This state-dependent gain control provides the exact mathematical mechanism that allows the gradient descent optimizer to easily sculpt the strict spectral gap and robust normal hyperbolicity observed in our RDNN models.

### E.4 Mechanism of Emergent Low-Rank Connectivity

The inverse scaling of the gradients by the dynamic denominator $(\boldsymbol{\eta} + \mathbf{G}_{t-1})$ acts as an activity-dependent implicit regularizer during gradient descent (Gunasekar et al., 2017; Woodworth et al., 2020).

- **Gradient Attenuation in High-Activity Regimes:** When the network is highly active along certain directions in the state space, the inhibitory pool $\mathbf{G}_{t-1}$ is driven strongly via $\mathbf{W}f(\mathbf{R}_{t-1})$, resulting in large values of $G_{t-1,j}$. Consequently, the gradient updates for the corresponding rows $J_{j,\cdot}$ are heavily suppressed by the factor $1/G_{t-1,j}$.

- **Selective Parameter Updates:** Weight updates are concentrated almost exclusively on neurons that are marginally active (where $G_{t-1,j}$ is small but $R_{t-1,k}$ is non-zero). This creates a competitive winner-take-all environment in the parameter space.

- **Implicit Nuclear Norm Minimization:** In deep learning theory, such multiplicative or gated update rules are known to exhibit an implicit bias toward sparsity and low-rank structures, similar to minimizing the nuclear norm of the weight matrices (Arora et al., 2019). By restricting the directions in parameter space that receive significant gradient updates, the network naturally collapses the recurrent weights $\mathbf{J}$ and $\mathbf{W}$ onto a low-rank subspace (as verified empirically in Table 1 and Figure 3**A**), without requiring explicit structural constraints.

### E.5 Optimization and Dynamical Landscapes of Ablated Networks

#### E.5.1 Subtractive Normalization and Attractor Shattering

We now mathematically analyze why the Subtractive Network sustains a stable ring attractor in the autonomous memory-guided saccade task but shatters into discrete point attractors under the input-driven angular velocity integration task, and why it consistently exhibits an elevated effective rank.

In the discrete-time formulation, the Jacobian of the Subtractive Network at step $t$ is given by:

$$\mathcal{J}_{\text{sub}}(t) = \boldsymbol{\Theta}_t \left[\text{diag}(\mathbf{1} - \boldsymbol{\alpha}_R) + \text{diag}(\boldsymbol{\alpha}_R)\mathbf{J}\,\text{diag}\left(f'(\mathbf{R}_{t-1})\right)\right], \tag{A-20}$$

where $\boldsymbol{\Theta}_t = \text{diag}\left(\Theta\left((\mathbf{1} - \boldsymbol{\alpha}_R) \odot \mathbf{R}_{t-1} + \boldsymbol{\alpha}_R \odot (\mathbf{J}f(\mathbf{R}_{t-1}) + \mathbf{I}_t - \mathbf{G}_{t-1})\right)\right) \in \{0,1\}^{H \times H}$ is a diagonal gating matrix defined by the Heaviside step function $\Theta(\cdot)$ originating from the state rectification boundary.

A continuous ring attractor requires a smooth 1-parameter family of fixed points satisfying the marginal stability condition, where the maximum eigenvalue modulus of the Jacobian is strictly bounded at unity ($|\lambda|_{\max} \approx 1$).

1. **Autonomous Regime ($\mathbf{I}_t = \mathbf{0}$):** The gating matrix $\boldsymbol{\Theta}_t$ remains static and spatially uniform over the persistent representation. Under these zero-input conditions, the network only needs to satisfy the marginal stability condition at a single unperturbed operating point. This is easily achieved through standard linear balancing of recurrent excitation and subtraction, preserving the ring topology.

2. **Input-Driven Regime ($\mathbf{I}_t \neq \mathbf{0}$):** As the external velocity input $\mathbf{I}_t$ continuously fluctuates over time, the term within the Heaviside function is directly modulated. This forces the diagonal gating matrix $\boldsymbol{\Theta}_t$ to undergo discrete, discontinuous state-switching. Consequently, as external inputs toggle the binary states in $\boldsymbol{\Theta}_t$, the effective recurrent connectivity undergoes abrupt structural changes. Without a dynamic divisor to smoothly normalize the Jacobian, its eigenvalues experience uncompensated discrete jumps. This mathematically violates the marginal stability constraint, triggering a sequence of saddle-node bifurcations that shatter the smooth ring manifold into discrete, highly stable point-attractor basins (sinks) separated by saddles (Figure 4**A**).

To understand why the Subtractive Network maintains a consistently higher effective rank than the RDNN (as illustrated in Figure 5 and Table A1), we analyze the BPTT gradient updates of the subtractive recurrent

weights. Denoting the local loss at step $t$ by $\mathcal{L}_t$, the direct gradient of $\mathcal{L}_t$ with respect to the subtractive recurrent weights $J_{jk}$ is given by:

$$\frac{\partial \mathcal{L}_t}{\partial J_{jk}} = e_{t,j} \cdot \alpha_{R,j} \cdot \Theta_{t,j} \cdot f(R_{t-1,k}), \tag{A-21}$$

where $e_{t,j} = \frac{\partial \mathcal{L}_t}{\partial R_{t,j}}$ is the backpropagated error, and

$$\Theta_{t,j} = \Theta \left( (1 - \alpha_{R,j}) R_{t-1,j} + \alpha_{R,j} \left( \mathbf{J}_{j,\cdot} f(\mathbf{R}_{t-1}) + I_{t,j} - G_{t-1,j} \right) \right) \in \{0, 1\} \tag{A-22}$$

is the binary activation state of neuron $j$ after rectification.

Contrast this with the gradient updates in the RDNN, where the updates are scaled continuously by the multiplicative divisor:

$$\frac{\partial \mathcal{L}_t}{\partial J_{jk}} \propto \frac{\alpha_{R,j}}{\eta_j + G_{t-1,j}}. \tag{A-23}$$

In the RDNN, the dynamic denominator $(\eta_j + G_{t-1,j})$ acts as an activity-dependent spectral compressor. When the network is active, $G_{t-1,j}$ is large, attenuating the weight updates along those dimensions and restricting parameter changes to a sparse, low-rank subspace.

In the Subtractive Network, however, the additive inhibition $G_{t-1,j}$ only modulates the gradient via the binary gate $\Theta_{t,j}$. For all active units along the persistent representation ($\Theta_{t,j} = 1$), the magnitude of the gradient update is completely decoupled from the scale of the inhibition $G_{t-1,j}$. Without this continuous multiplicative scaling, the optimization path lacks the implicit regularization bias. Consequently, the weight updates propagate across a wider, unconstrained subspace of $\mathbb{R}^{H \times H}$, leading to the elevated effective rank observed in our quantitative scaling analysis.

**BPTT Optimization and Dimensional Compensation** This gradient-level decoupling also elucidates the discrepancy between the topological fragmentation and the seemingly similar eigenvalue modulus spectra in the angular velocity integration task (Figure 4**A** and A14).

To minimize the integration loss over a long horizon $T$, BPTT must prevent gradient vanishing along the active trajectory by forcing the task-relevant eigenvalues to satisfy $|\lambda| \approx 1$. In the RDNN, the dynamic denominator $\boldsymbol{\eta} + \mathbf{G}$ acts as a spectral compressor, driving orthogonal (normal) directions to $|\lambda_{\text{normal}}| \to 0$ (normal hyperbolicity, see Appendix E.3). This confines the slow modes strictly to the low-dimensional tangent space of the ring, allowing saddles to act as efficient *slow channels* for continuous integration (Sussillo & Barak, 2013).

In the Subtractive Network, the continuous manifold shatters into a high density of discrete stable point-attractors (sinks). To bridge these discrete wells and prevent the state representation from decaying during input-driven transitions, BPTT is forced to preserve information across a much wider subspace. While the post-update hard-thresholding of the ReLU function still enforces structural zero eigenvalues for strictly silent neurons, as $\boldsymbol{\Theta}_t$ zeros out their entire corresponding rows in the Jacobian, accounting for the visible density peak near $|\lambda| \approx 0$ (Glorot et al., 2011), the subtractive gradient lacks the multiplicative scaling to dynamically contract the active normal dimensions. Therefore, the optimizer cannot efficiently suppress non-tangent directions among the active subpopulation (Arora et al., 2019; Gunasekar et al., 2017). Consequently, while both networks exhibit a prominent density peak near $|\lambda| \approx 1$ to satisfy the BPTT memory constraint, the Subtractive Network achieves this by recruiting a higher number of physical dimensions to construct redundant slow pathways across the fragmented landscape. This dimensional compensation is mathematically manifested as the significantly elevated effective rank observed in Figure 5.

### E.5.2 Spectral Alignment and Hyper-Compression in Low-Rank RDNN

We address why the trained Explicit Low-Rank RDNN exhibits an effective rank that is significantly lower than both the permitted bottleneck rank $r$ and the unconstrained RDNN's effective rank.

When the recurrent weights are factorized as $\mathbf{J} = \mathbf{J}_1 \mathbf{J}_2$ and $\mathbf{W} = \mathbf{W}_1 \mathbf{W}_2$, the gradients of the BPTT loss $\mathcal{L}$ with respect to the factorized matrices are:

$$\nabla_{\mathbf{J}_1} \mathcal{L} = (\nabla_{\mathbf{J}} \mathcal{L}) \mathbf{J}_2^T, \quad \nabla_{\mathbf{J}_2} \mathcal{L} = \mathbf{J}_1^T (\nabla_{\mathbf{J}} \mathcal{L}). \tag{A-24}$$

Under gradient descent, this coupled product formulation induces **strong spectral alignment** (or co-adaptation) between the column space of $\mathbf{J}_1$ and the row space of $\mathbf{J}_2$. In deep matrix factorization theory (Arora et al., 2019; Gunasekar et al., 2017), this multiplicative gradient flow acts as an accelerator for singular value decay. Specifically, the singular values of the product $\mathbf{J}_1 \mathbf{J}_2$ decay exponentially fast compared to those of a flat matrix $\mathbf{J}$ trained directly. This is because the updates to the dominant singular vectors are mutually reinforced by the transpose factors:

$$\Delta \sigma_i(\mathbf{J}) \propto \sigma_i(\mathbf{J}_1) \cdot \sigma_i(\mathbf{J}_2). \tag{A-25}$$

This factorized gradient dynamics acts as a powerful multiplier to the implicit divisive regularizer. As a result, the network undergoes an extreme hyper-compression, discarding almost all available dimensions of the bottleneck $r$ and concentrating 99% of its spectral energy into an exceptionally tight, low-dimensional subspace of rank $\approx 2 \sim 4$.

### E.5.3  High-Rank Convergence Anomaly in Explicit Low-Rank Factorization

For the Explicit Low-Rank RDNN, the gradient of the loss with respect to the factorized matrices (e.g., $\mathbf{J}_1$ and $\mathbf{J}_2$) is given by:

$$\frac{\partial \mathcal{L}}{\partial \mathbf{J}_1} = \frac{\partial \mathcal{L}}{\partial \mathbf{J}} \mathbf{J}_2^T, \quad \frac{\partial \mathcal{L}}{\partial \mathbf{J}_2} = \mathbf{J}_1^T \frac{\partial \mathcal{L}}{\partial \mathbf{J}}. \tag{A-26}$$

While this factorization guarantees $\text{rank}(\mathbf{J}) \leq r$, it introduces severe non-convexities into the optimization landscape:

- **Scaling Symmetries and Flat Valleys**: For any invertible matrix $\mathbf{P} \in \mathbb{R}^{r \times r}$, the transformation $(\mathbf{J}_1 \mathbf{P}, \mathbf{P}^{-1} \mathbf{J}_2)$ leaves the product $\mathbf{J}$ invariant. This continuous symmetry creates infinitely many flat directions (valleys) in the loss landscape. As dictated by the cross-dependent gradients in Eq. A-26, drifting along these valleys can induce severe scale imbalances between the factors (e.g., $\|\mathbf{J}_2\| \to 0$). When this occurs, the gradient for the other factor ($\frac{\partial \mathcal{L}}{\partial \mathbf{J}_1} \propto \mathbf{J}_2^T$) vanishes and stalls the optimization process (Li et al., 2018; Zhao et al., 2026).

- **The High-Rank Convergence Anomaly**: When the permitted explicit bottleneck rank $r$ is unnecessarily large (e.g., $r = 32$) relative to the task's true low-dimensional manifold (which we showed requires only rank $\approx 2 \sim 4$), the number of redundant symmetries and saddle points scales quadratically with $r$ (Li et al., 2019; Ge et al., 2017). This over-parameterization of the bottleneck introduces a vast number of degenerate saddle points and flat plateaus, severely hindering convergence (Xiong et al., 2024) (as observed in the high variance of $r = 32$ in Figure 3**B**). A tight bottleneck ($r = 8$) minimizes these redundant degrees of freedom, smoothing out the optimization landscape and enabling rapid convergence.

### E.5.4  Linear Parameterization and the Absence of Anomaly in Unconstrained RDNN

We analyze why the unconstrained (full-rank) RDNN does not suffer from the convergence anomaly as its hidden size $H$ increases, unlike the explicitly factorized network.

In the unconstrained RDNN, the parameters are represented directly by the flat recurrent matrices $\mathbf{J}, \mathbf{W} \in \mathbb{R}^{H \times H}$, avoiding any product-form parameterization.

1. **Absence of Scaling Symmetries**: The parameter-to-weight mapping is a trivial identity. Thus, the unconstrained formulation does not introduce the continuous non-convex scaling symmetries $(\mathbf{J}_1 \mathbf{P}, \mathbf{P}^{-1} \mathbf{J}_2)$ that create flat valleys and zero-gradient directions (Dinh et al., 2017; Li et al., 2019).

2. **Landscape Smoothing via Over-Parameterization**: In deep learning theory, increasing the dimensionality $H$ of a directly parameterized weight matrix $\mathbf{J}$ leads to standard over-parameterization. This over-parameterization mathematically smoothes the loss landscape by creating highly connected, high-dimensional paths that eliminate bad local minima and degenerate saddle points (Du et al., 2019; Cooper, 2018). Consequently, larger unconstrained RDNNs (e.g., $H = 256$) train more rapidly and stably than smaller ones, whereas larger explicit bottlenecks $r$ in factorized networks exacerbate the non-convex factorization pathology.

## F    Extended Slow Manifold Analysis

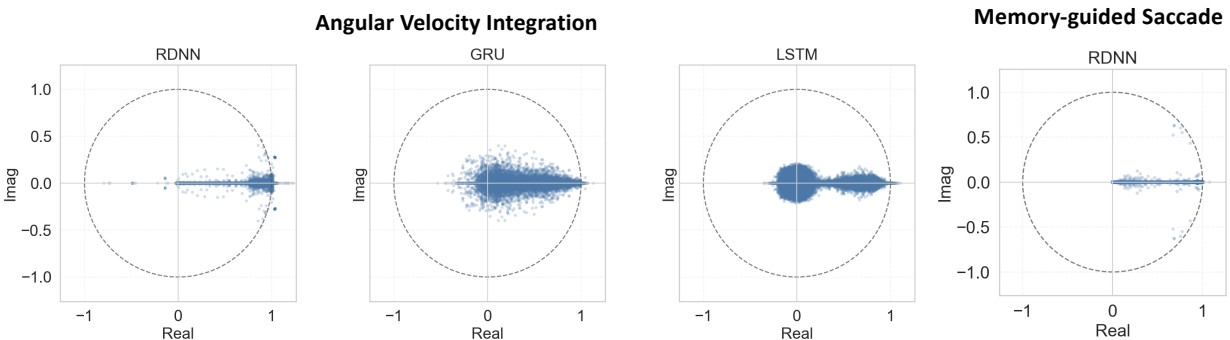

Figure A5: Complex eigenvalue spectra of the Jacobians evaluated on the slow manifolds. In the angular velocity integration task, the RDNN exhibits a highly structured, tripartite spectrum consisting of a dense cluster along the real axis (governing fast dissipation), off-axis conjugate pairs in the interior (stabilizing transient transitions), and boundary conjugate pairs near the unit circle (facilitating continuous rotational dynamics). In the memory-guided saccade task, the RDNN suppresses these imaginary components, aligning its slow modes along the real axis to maintain static memory. In contrast, GRU and LSTM exhibit diffuse spectral clouds lacking a clear spectral gap or structured rotational modes.

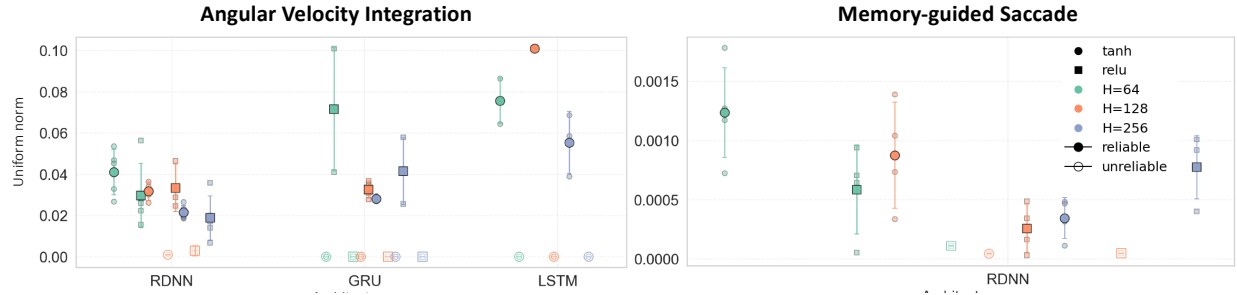

Figure A6: The uniform norm quantifies the maximum drift speed along the manifold; a lower value indicates a flatter energy landscape and a higher-fidelity continuous attractor. The RDNN maintains a substantially lower uniform norm than the baselines in the integration task (left), and achieves near-zero drift (on the order of $10^{-3}$ to $10^{-4}$) in the autonomous saccade task (right). Markers indicate the mean across reliable instances, with error bars denoting the standard deviation.

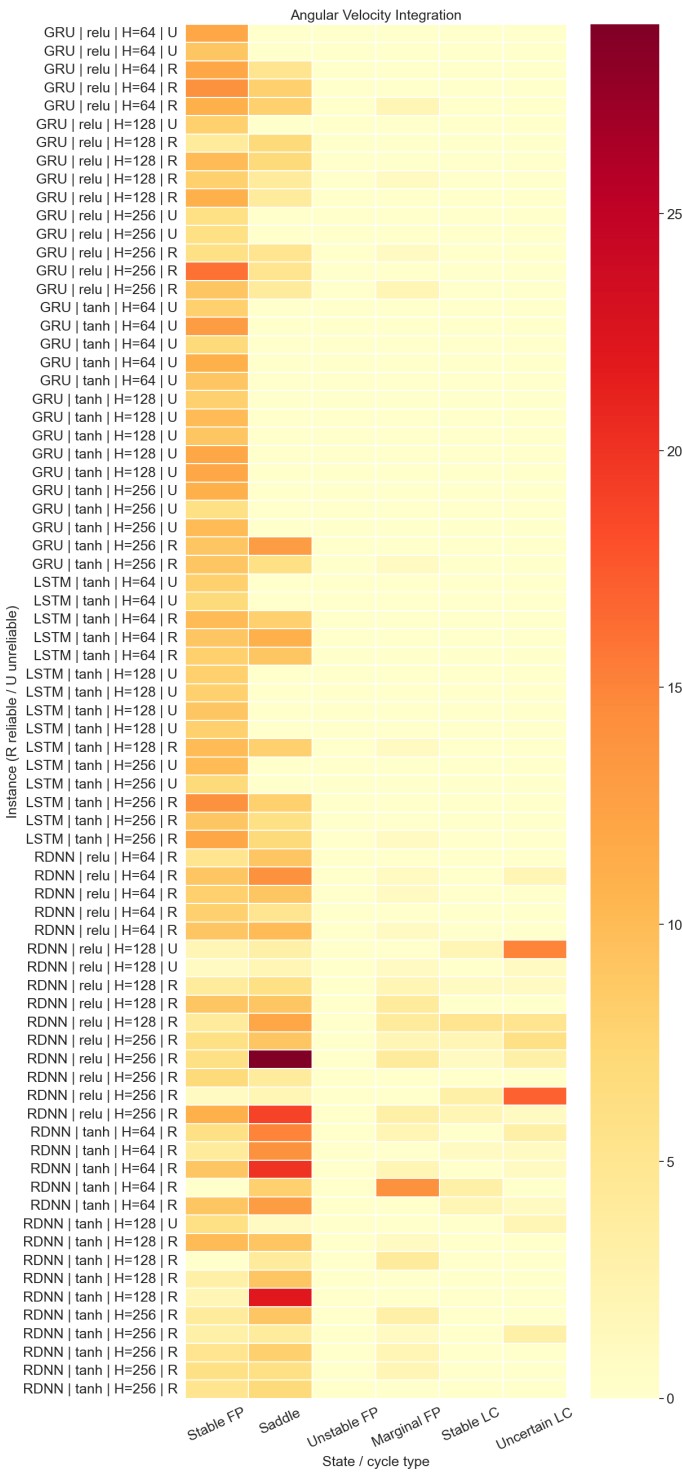

Figure A7: Instance-level equilibrium classification for the angular velocity integration task. The heatmap displays the absolute count of different dynamical states (fixed points and limit cycles) for each trained instance (seed) across various configurations. While GRU and LSTM instances are strictly dominated by alternating stable fixed points and saddles (indicating discretized state spaces), RDNN instances exhibit a more fluid topology, frequently developing marginal fixed points and stable limit cycles to support continuous integration.

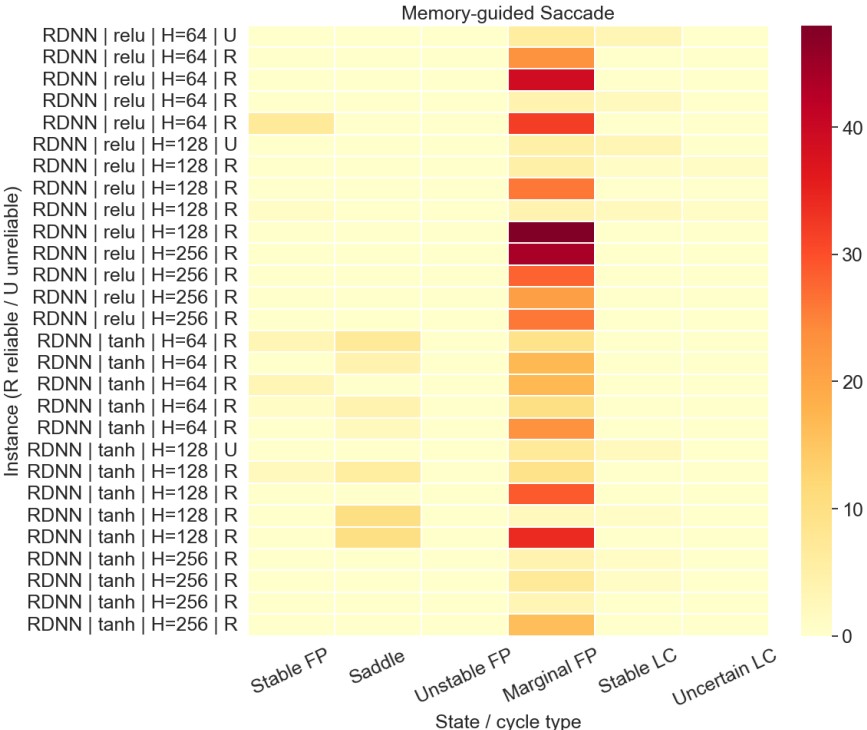

Figure A8: Instance-level equilibrium classification for the memory-guided saccade task. The heatmap details the dynamical states of the RDNN across all trained seeds and configurations. The network consistently converges to solutions overwhelmingly dominated by marginal fixed points. This instance-level consistency confirms that divisive normalization provides a strong inductive bias for forming approximate continuous attractors for autonomous memory maintenance.

## G  Extended Analysis of Explicit Low-Rank Parameterization

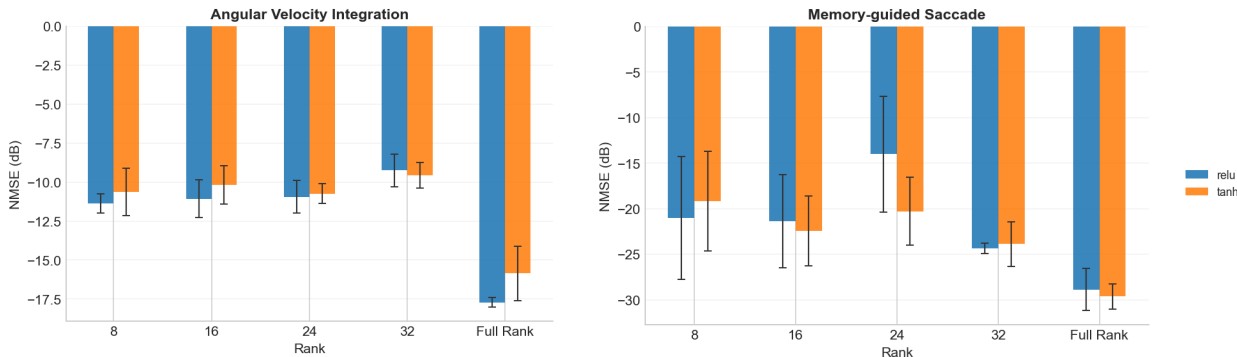

Figure A9: Test NMSE of explicitly factorized RDNNs. Normalized Mean Squared Error (NMSE, in dB) on the test set of 1024 trajectories for the $H = 128$ RDNN with explicitly factorized recurrent weights ($\mathbf{J}$ and $\mathbf{W}$) across bottleneck ranks $r \in \{8, 16, 24, 32\}$, compared with the unconstrained Full-Rank RDNN. Left: Angular Velocity Integration task. Right: Memory-guided Saccade task. Error bars denote the standard deviation across different random seeds. Even under an extremely tight bottleneck (e.g., $r = 8$), the RDNN maintains highly competitive task performance, demonstrating the intrinsic low-dimensional nature of the continuous working memory representation.

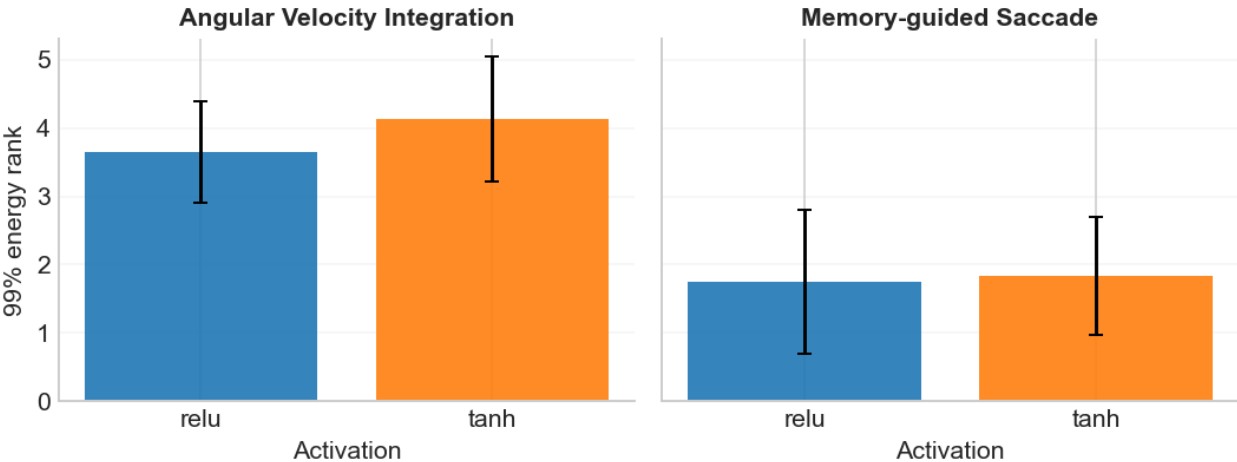

Figure A10: Self-compressed effective rank of explicitly factorized RDNNs. The post-training 99% energy effective rank of the $H = 128$ RDNN with explicitly factorized weights across activation functions (relu and tanh). Left: Angular Velocity Integration task. Right: Memory-guided Saccade task. Despite having an explicit parameter bottleneck of $r \in \{8, 16, 24, 32\}$, the trained models consistently compress their effective dimensionality to approximately $3.5 \sim 4.0$ (Angular Velocity Integration) and $\sim 2.0$ (Memory-guided Saccade), indicating that the network actively discards redundant dimensions to align with the task's low-dimensional manifold.

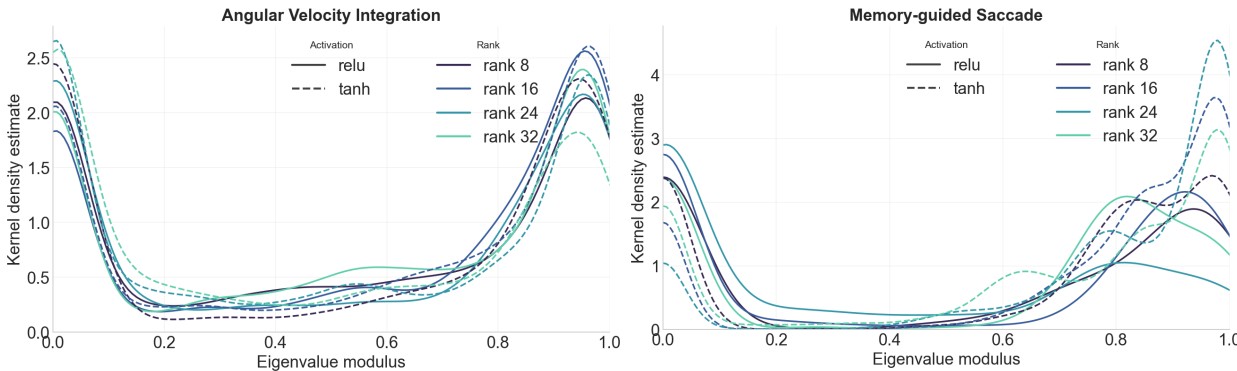

Figure A11: Jacobian eigenvalue modulus density of explicit low-rank RDNNs. Kernel density estimate (KDE) of the eigenvalue modulus ($|\lambda|$) evaluated on the slow manifold for the $H = 128$ explicitly factorized RDNNs across ranks $r \in \{8, 16, 24, 32\}$ and activations (relu as dashed, tanh as solid lines). Left: Angular Velocity Integration task. Right: Memory-guided Saccade task. While the bimodal structure (representing slow-fast separation) is largely preserved, some configurations in the saccade task exhibit minor eigenvalue leakage in the intermediate region, reflecting a slight degradation of the strict spectral gap due to explicit low-rank constraints.

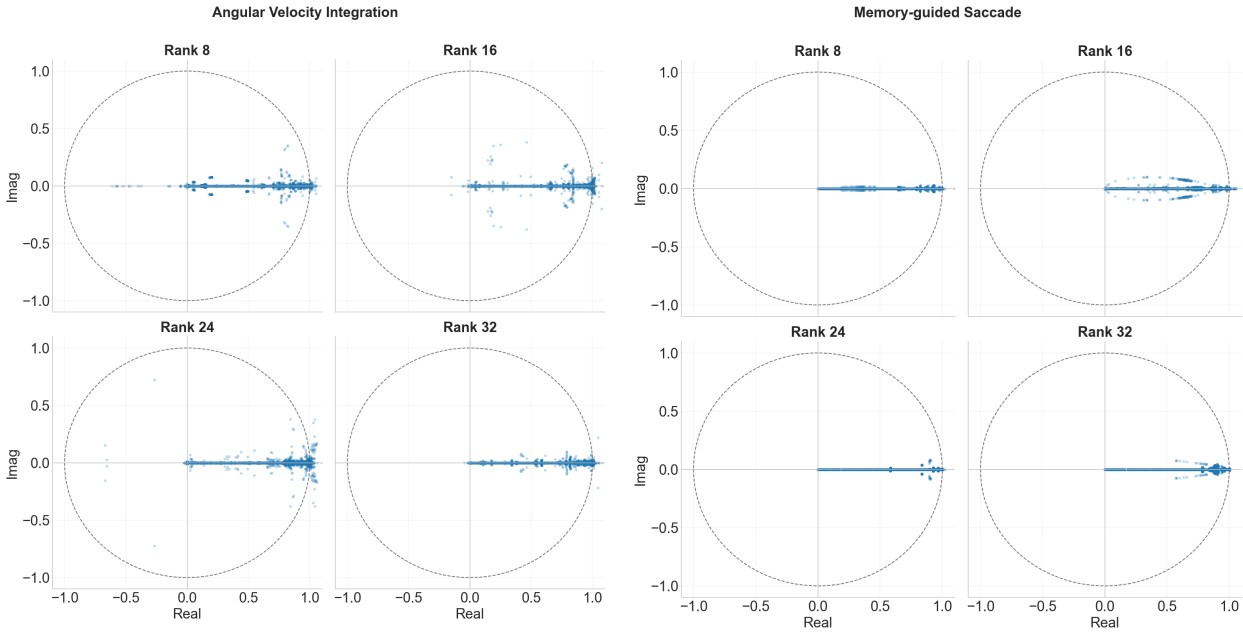

Figure A12: Complex plane eigenvalue spectra of explicit low-rank RDNNs. Scatter plots of the complex Jacobian eigenvalues evaluated on the slow manifold across different bottleneck ranks $r \in \{8, 16, 24, 32\}$ (rows represent different explicit ranks). Left: Angular Velocity Integration task. Right: Memory-guided Saccade task. The tripartite spectral structure in the integration task (real-axis clustering, interior and boundary conjugate pairs) and the strict real-axis alignment in the saccade task are largely preserved with explicit factorization, though higher ranks introduce minor off-axis scattering on the saccade manifold.

## H    Extended Ablation Analysis

Table A1: The 99% energy effective rank (mean ± std over multiple seeds) across RDNN and Subtractive Network.

| Model | Activation | Angular Velocity Integration | | | Memory-guided Saccade | | |
|---|---|---|---|---|---|---|---|
| | | $H = 64$ | $H = 128$ | $H = 256$ | $H = 64$ | $H = 128$ | $H = 256$ |
| RDNN | relu | $16.8 \pm 2.4$ | $24.0 \pm 2.5$ | $38.0 \pm 6.7$ | $19.2 \pm 2.5$ | $27.0 \pm 7.2$ | $31.0 \pm 5.0$ |
| | tanh | $17.2 \pm 1.9$ | $22.6 \pm 2.6$ | $35.4 \pm 5.3$ | $16.0 \pm 2.0$ | $22.6 \pm 3.5$ | $28.3 \pm 5.6$ |
| Subtractive Network | relu | $24.0 \pm 1.0$ | $37.6 \pm 1.5$ | $51.0 \pm 2.7$ | $24.0 \pm 2.5$ | $29.4 \pm 4.1$ | $40.25 \pm 3.9$ |
| | tanh | $25.2 \pm 2.3$ | $40.2 \pm 3.6$ | $52.2 \pm 3.3$ | $21.0 \pm 2.2$ | $29.6 \pm 2.2$ | $37.0 \pm 5.7$ |

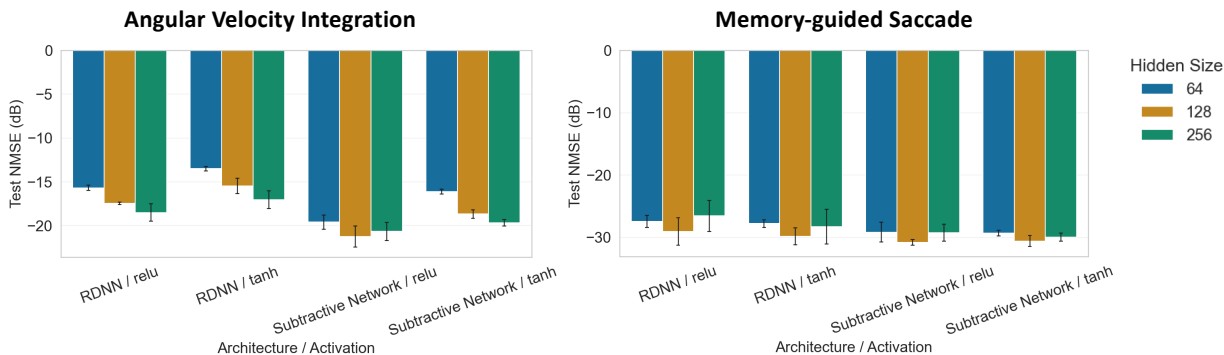

Figure A13: Performance comparison (Test NMSE) of divisive (RDNN) and subtractive networks. Test set Normalized Mean Squared Error (NMSE, in dB) of 1024 trajectories on both tasks across different hidden sizes and activation functions. While the Subtractive Network achieves highly competitive or marginally lower test NMSE compared to the RDNN, its underlying dynamical representations with external inputs are highly discretized (as shown in Figures 4 and A16), contrasting with the fluid, low-rank manifolds learned by the RDNN. Error bars denote the standard deviation across different random seeds.

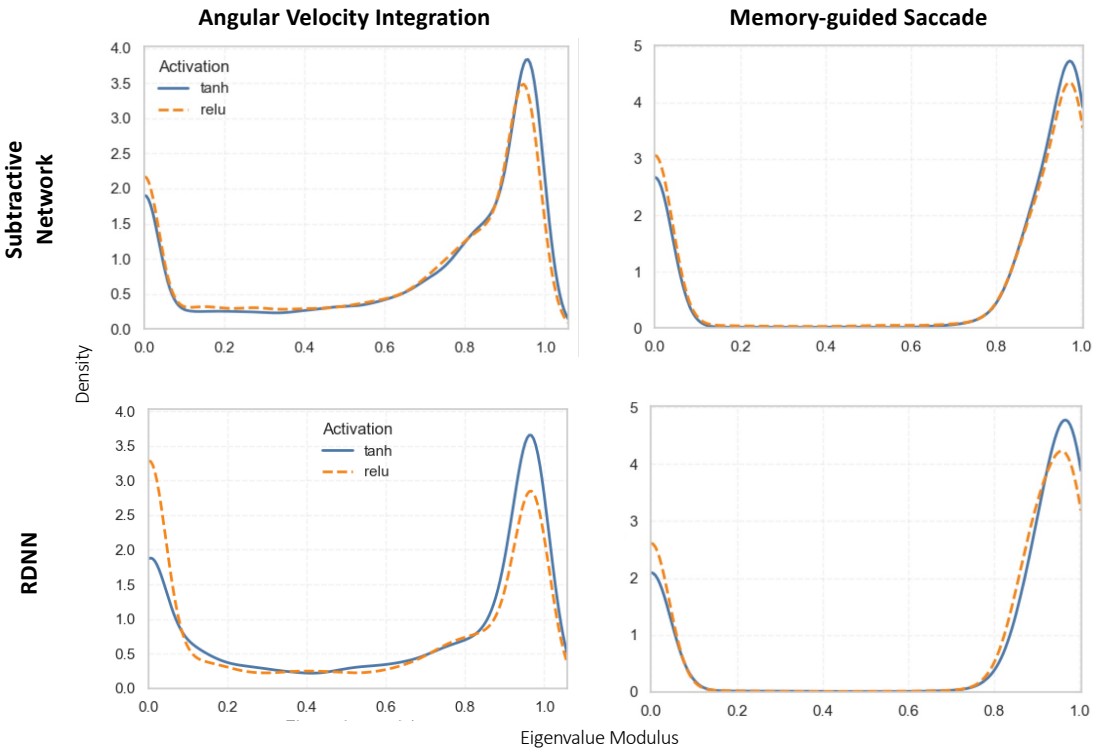

Figure A14: Eigenvalue modulus density spectra of divisive and subtractive networks. Kernel density estimates (KDE) of the Jacobian eigenvalue moduli ($|\lambda|$) evaluated on the slow manifolds. Left (Angular Velocity Integration): Both networks exhibit a prominent peak near $|\lambda| \approx 1$ to satisfy the BPTT memory constraint for continuous integration. Right (Memory-guided Saccade): Both networks display nearly identical, sharply separated bimodal spectra with a strict spectral gap, confirming that both mechanisms can achieve marginal stability in the absence of external input fluctuations.

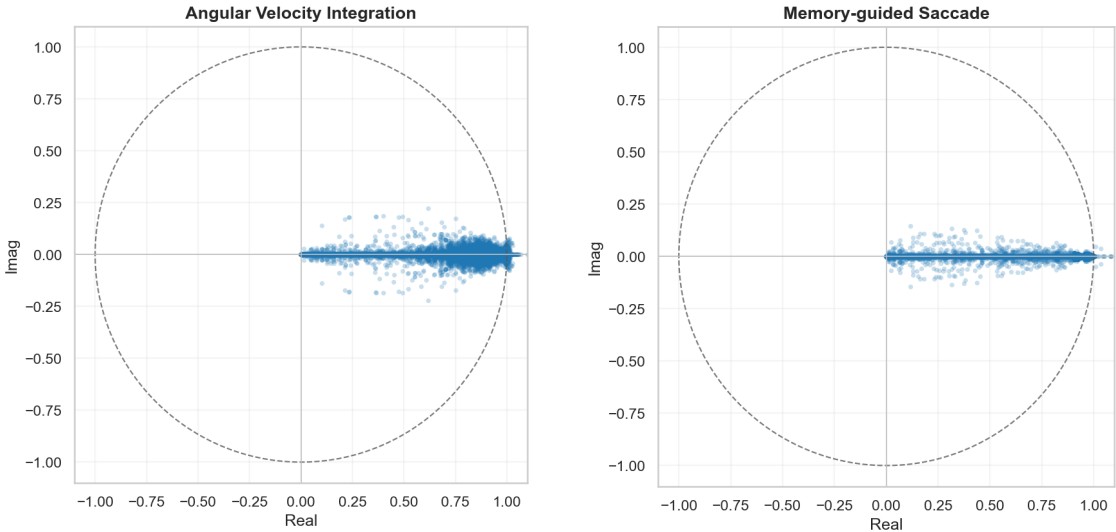

Figure A15: Complex plane eigenvalue spectra of the Subtractive Network. Scatter plots of the complex Jacobian eigenvalues evaluated on the slow manifold of the Subtractive Network. Left: Angular Velocity Integration task. Right: Memory-guided Saccade task. Unlike the RDNN (cf. Figure A5), the Subtractive Network's eigenvalues remain heavily concentrated along the real axis even during continuous integration (left), lacking the boundary-hugging complex conjugate pairs required for smooth, marginally stable rotational dynamics.

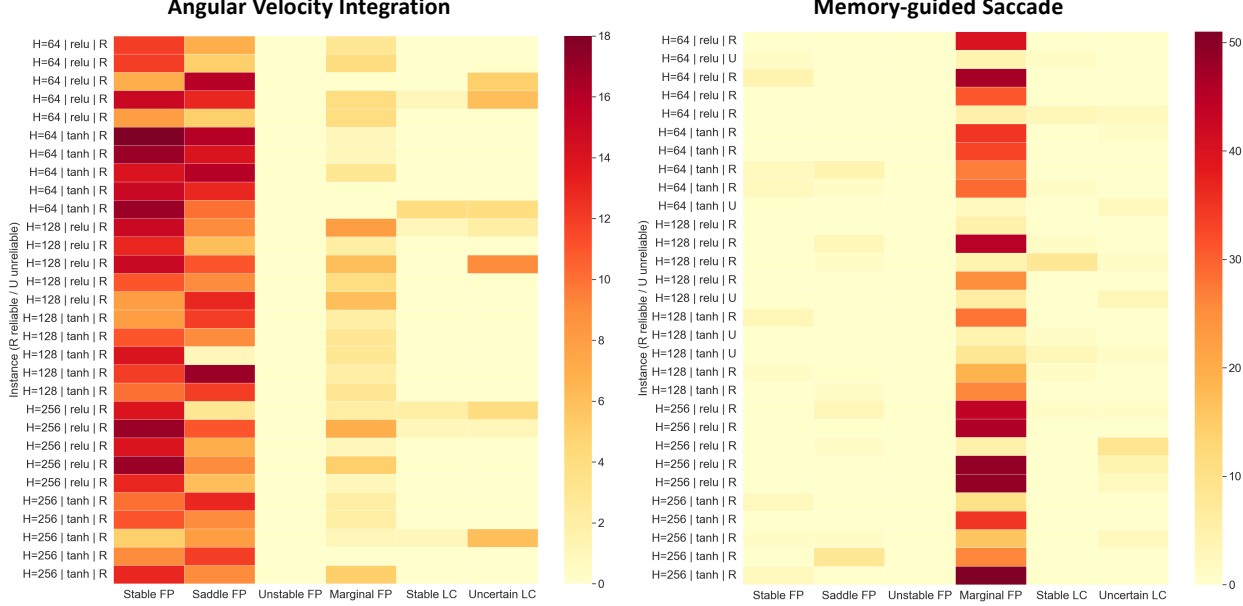

Figure A16: The heatmaps display the absolute count of different dynamical states (fixed points and limit cycles) of the Subtractive Network across all trained random seeds and configurations for the angular velocity integration task (left) and the memory-guided saccade task (right). In the integration task, the Subtractive Network consistently converges to a discretized state space dominated by stable fixed points and saddles across every single seed, demonstrating that attractor shattering is a structural consequence of additive subtraction. In the autonomous saccade task (right), it robustly maintains marginal fixed points, matching the qualitative behavior of the RDNN under zero-input conditions.

