# OpenReview forum: "Divisive Normalization Shapes Low-Rank Slow Manifolds for Continuous Working Memory"
_TMLR — Under review for TMLR_

### Review · Reviewer_YvNQ · 2026-07-13

**Summary Of Contributions:**

This paper introduces a new method for improving recurrent networks using divisive normalization, called Recurrent Divisive Normalization Network (RDNN). RDNN aims for continuous working-memory tasks. Specifically, this method combines an excitatory recurrent state with a dynamically coupled inhibitory state that divisively modulates the recurrent drive. The authors evaluate the model on angular-velocity integration and memory-guided saccade tasks and compare it with GRU and LSTM baselines. The authors also conduct a comprehensive analysis of model performance in terms of fixed points, limit cycles, Jacobian eigenspectra, manifold drift, and the effective rank of recurrent weight matrices.

## Strengths:

- This paper introduces an interesting biologically inspired mechanism for improving recurrent dynamics, using divisive normalization as an inductive bias for learning continuous dynamical representations.

- Beyond prediction accuracy, the paper explicitly focuses on the continuity and geometric structure of the learned state-space dynamics.

- The paper provides extensive empirical analyses of the learned model geometry, including fixed-point structure, Jacobian eigenspectra, effective rank analysis, etc.

- This paper is generally well-presented and well-organized.


## Weaknesses:

I have several major concerns regarding this paper. Please see below.

- First, the baseline comparison is weak. The proposed method aims for learning continuity. I would not be surprised that GRU and LSTM do not work well as baselines. The authors need to consider more advanced neural architectures for learning continuous-time dynamics, such as NeuralODEs [1] and some transformer-based architectures [2]. Also, it would be good to report the model parameters in terms of the model performance.

Refs:

[1] Chen, Ricky TQ, et al. "Neural ordinary differential equations." Advances in neural information processing systems 31 (2018).

[2] Wu, Haixu, et al. "Autoformer: Decomposition transformers with auto-correlation for long-term series forecasting." Advances in neural information processing systems 34 (2021): 22419-22430.


- Second, the experimental scope is weak. The current experiments only consider one-dimensional variables. It is difficult to tell whether the conclusions and claims of this paper can be generalized to more complex dynamical systems. It would be better for the authors to consider high-dimensional variables/systems, coupled systems, etc. Also, for the experimental setups, it would be good to test continuity and accuracy in long-horizon prediction and out-of-distribution performance tasks.

- Third, it would be better to clarify the difference and novelty of the proposed method compared to ORGaNICs and Rawat et al. (2024) in the Related Works. Also, why don’t you consider those two methods as baselines?

- Also, I am a bit curious how you compute the effective rank. It looks like that the authors compute the effective rank in a concatenated way for recurrent weights. Is that the correct way to do so?

**Additional Comments:**

N/A.

**Audience:**

Yes

**Audience Explanation:**

This paper is related to neural network architectures, biology-inspired inductive bias, and dynamics modeling. The relevant readers might be interested in this paper.

**Broader Impact Concerns:**

N/A.

**Claims And Evidence:**

No

**Claims Explanation:**

Many of the claims are problematic.

For example, the claims are based on the current experiments and baseline comparisons. However, I didn't find the current experiments sufficiently convincing.

Second, some claims like "inverse gradient scaling from divisive normalization acts as an implicit low-rank regularizer" are not rigorously correct. The rescaling is not equivalent to minimizing rank, stable rank, or singular-value entropy, etc. A more rigorous argument would need to show that the proposed dynamics systematically produce differential decay of singular modes, or that the optimization dynamics favor a low-rank solution under clearly stated assumptions. The current derivation appears closer to a local gradient-scaling argument. That may explain reduced update magnitude in certain activity regimes, but it does not by itself establish rank compression.

**Requested Changes:**

Following the weaknesses part, I would suggest:

- Include more advanced baselines.
- Include more complex experimental setups.
- Clarify the novelty of the proposed method.
- Tone down some of the claims and carefully verify the computations and formulations of the metrics used in the analysis.

---

### Review · Reviewer_WuNF · 2026-07-18

**Summary Of Contributions:**

This paper considers a novel biologically-inspired RDNN architecture for continuous memory by integrating divisive normalization (DN) into an RNN circuit. The main novelty in the architecture is the combination of existing tools with some smaller modifications compared to previous works, such as ORGaNIC. In comparison with GRU/LSTM, the authors show using a dynamical-systems viewpoint that the RDNN architecture stores a continuous value as a smooth nearly flat "ring" of equally stable states, whereas competitors discretize the continuous spectrum (Section 4). Moreover, Section 5 provides evidence that divisive normalization acts as an implicit regularizer and compresses the network into few dimensions, linking a biological mechanism to a phenomenon in ML theory. Finally, the authors provide an ablation (Section 6) by replacing the divisive mechanism with a subtractive one, which demonstrates that the subtraction is enough for holding memory, but division is required to keep it working during changing inputs.

**Additional Comments:**

- there is a typo in equation A-6 in the appendix ($E[y^2]$), but for this task the target is on the unit circle, so it can be reduced to MSE.

**Audience:**

Yes

**Audience Explanation:**

The topic of this paper is relevant as it connects a biological mechanism (DN) to an active question in ML theory, providing insights into how and why trained networks implicitly compress themselves into low-dimensional representations.

**Claims And Evidence:**

No

**Claims Explanation:**

The paper is overall well written and detailed. However, there are some central claims that are not sufficiently supported:

- the paper's main motivation is against CANs, but while the authors show that the RDNN architecture provides a robust slow manifold, they never directly compare its drift-under-perturbation against the CAN, which leaves the claim "overcome the fine-tuning problem" not sufficiently supported
- In Section 4, it is argued that the slow-manifold topology is a result from DN. However, it is not investigated whether other DN-type architectures, e.g. ORGaNIC, provide a similar behavior, or if this finding is specific to the architecture provided here
- in some settings, e.g. the saccade task, GRU and LSTM do not converge and cannot act as a reliable baseline, and thus there is no baseline for comparison present for this task

**Requested Changes:**

- comparison against the CAN as outlined above
- comparison with other DN-type architectures as outlined above

---

### Review · Reviewer_t79d · 2026-07-19

**Summary Of Contributions:**

strengths
1. The paper presents a biologically motivated recurrent architecture based on divisive normalization and provides an interesting perspective on continuous working memory through the lens of dynamical systems.

2. The work offers a comprehensive mechanistic analysis, including slow-manifold dynamics, effective-rank evolution, spectral properties, and ablation studies, providing valuable insights beyond task-level performance.

3. The experimental results consistently demonstrate the effectiveness of the proposed method on canonical continuous working memory tasks, and the presentation is generally clear with thorough empirical evaluations and theoretical discussions.

weaknesses
1. The paper's motivation and positioning with respect to prior work could be further strengthened. Recent studies (e.g., ORGaNICs) have already incorporated divisive normalization into recurrent neural networks for working memory modeling and have analyzed their stability and dynamical properties. While this work provides a more in-depth investigation from the perspectives of slow manifolds, low-rank representations, and optimization dynamics, it does not sufficiently clarify the specific limitations of these existing approaches or clearly articulate the fundamental differences between RDNN and previous divisive-normalization-based models in terms of architectural design or underlying mechanisms. A more explicit discussion of the core scientific problem addressed by this work and its unique contributions beyond prior studies would further strengthen the paper's motivation and novelty.

2. The proposed method is mainly compared against GRU and LSTM baselines, but lacks comparisons with more recent continuous-time neural models or state-space models (e.g., Neural ODEs, Liquid Neural Networks, or Mamba-style continuous state-space models). As a result, it remains difficult to fully assess the advantages of the proposed approach relative to current state-of-the-art continuous-time architectures.

3. RDNN introduces an additional inhibitory neural state together with several learnable parameters, resulting in a more complex architecture than conventional recurrent networks. However, the paper does not provide a systematic analysis of the resulting parameter count, computational cost, or inference efficiency, making it difficult to evaluate the practical deployment overhead of the proposed model. Moreover, although experiments are conducted across different random seeds and hidden dimensions, the sensitivity of the method to factors such as noise levels, hyperparameter choices, and network scale is only partially explored. Consequently, the robustness of the proposed approach under broader settings remains to be further validated.

4. The discussion highlights several promising application domains, including spatial navigation, robotics, and parameter-efficient continuous-time modeling. However, the empirical evaluation is currently limited to canonical low-dimensional synthetic tasks. While these benchmarks are appropriate for validating the proposed dynamical mechanisms, demonstrating the method on more realistic or application-oriented scenarios would provide stronger evidence for its practical utility and generalization capability.

**Audience:**

Yes

**Audience Explanation:**

The paper addresses biologically inspired recurrent neural networks and continuous working memory, which are topics of interest to researchers in machine learning, computational neuroscience, and dynamical systems.

**Broader Impact Concerns:**

This work is primarily a methodological study on biologically inspired recurrent neural networks and continuous working memory. I do not identify any significant ethical concerns that would require an additional Broader Impact Statement.

**Claims And Evidence:**

Yes

**Claims Explanation:**

The paper provides extensive empirical evaluations and dynamical analyses, including slow-manifold characterization, spectral analysis, effective-rank analysis, and ablation studies, which generally support the main claims regarding the proposed RDNN. The theoretical and experimental evidence is presented clearly and consistently. However, some broader claims about practical applicability and the advantages over existing continuous-time architectures are not yet fully substantiated, as the experiments are limited to canonical low-dimensional synthetic tasks and comparisons are primarily made against GRU and LSTM baselines. Additional evaluations on more realistic scenarios and stronger baselines would further strengthen the evidence supporting these claims.

**Requested Changes:**

1. Clarify the paper's novelty and positioning with respect to prior work. In particular, explicitly discuss the limitations of existing divisive-normalization-based recurrent models (e.g., ORGaNICs) and clearly articulate the fundamental architectural and mechanistic differences between RDNN and previous approaches.

2. Strengthen the experimental evaluation by including comparisons with more recent continuous-time neural models or state-space models (e.g., Neural ODEs, Liquid Neural Networks, or Mamba-style architectures). In addition, provide analyses of parameter count, computational complexity, inference efficiency, and robustness under different noise levels, hyperparameter settings, and network scales.

3. Provide stronger evidence for the claimed practical applicability by evaluating the proposed method on more realistic or application-oriented tasks beyond canonical low-dimensional synthetic benchmarks, particularly for the application domains discussed in the paper (e.g., spatial navigation or robotics).